# Personalized Online Federated Learning with Multiple Kernels

**Pouya M. Ghari**
University of California Irvine
pmollaeb@uci.edu

**Yanning Shen** *
University of California Irvine
yannings@uci.edu

## Abstract

Multi-kernel learning (MKL) exhibits well-documented performance in online non-linear function approximation. Federated learning enables a group of learners (called clients) to train an MKL model on the data distributed among clients to perform online non-linear function approximation. There are some challenges in online federated MKL that need to be addressed: i) Communication efficiency especially when a large number of kernels are considered ii) Heterogeneous data distribution among clients. The present paper develops an algorithmic framework to enable clients to communicate with the server to send their updates with affordable communication cost while clients employ a large dictionary of kernels. Utilizing random feature (RF) approximation, the present paper proposes scalable online federated MKL algorithm. We prove that using the proposed online federated MKL algorithm, each client enjoys sub-linear regret with respect to the RF approximation of its best kernel in hindsight, which indicates that the proposed algorithm can effectively deal with heterogeneity of the data distributed among clients. Experimental results on real datasets showcase the advantages of the proposed algorithm compared with other online federated kernel learning ones.

## 1 Introduction

Kernel learning exhibits well-documented performance in function approximation tasks, while providing theoretical guarantees associated with different performance metrics, see e.g. [48, 21, 38]. In some cases, a group of learners aims at collaborating to perform function approximation without revealing their data. To this end, federated learning has been emerged as a crucial learning paradigm by enabling a group of learners called clients to collaborate with each other by communicating with a central server to train a centralized model [33, 27, 12, 23]. Through this process, clients send model parameters and updates to the server without revealing their data. Upon receiving updates from clients, the server updates the model. Therefore, federated learning enables clients to perform kernel learning for function approximation. In this context, a server and clients collaborate with each other to learn the optimal kernel. Furthermore, in some practical cases, clients may need to perform the function approximation in an online fashion while they are collaborating with the server to learn the kernel. For example, consider the case where clients may not have enough memory to store data in batch. In addition, data samples may arrive in a sequential manner such that clients are not able to perform the function approximation in batch form. There are major challenges in performing online kernel learning in federated fashion that need to be addressed:

**Communication Efficiency:** Communication efficiency arises as a bottleneck in federated learning (see e.g. [25, 37, 19, 16]). Specifically, limited clients-to-server communication bandwidth restricts the number of parameters that can be sent from clients to the server.

**Heterogeneous Data:** The distribution of data observed by a client might be different from others

---

*Corresponding author

(see e.g. [44, 29, 6]). Thus, the optimal kernel that is aimed to be learned is different across clients.
**Computational Complexity:** Clients should be able to perform function approximation fast enough in order to make a decision in real-time. Therefore, the computational complexity of kernel learning methods should be affordable for clients.

Conventional online kernel learning approaches (see e.g. [21, 39]) suffer from 'curse of dimensionality' [3] in the sense that the number of parameters that should be learned increases with the number of observed data. This can make employing conventional online kernel learning approaches infeasible to perform online federated kernel learning since clients may be required to send a large number of parameter updates to the server while the available clients-to-server communication bandwidth is not enough for sending such information. Approximating kernels by finite-dimensional feature representations (e.g. Nyström method [49] and random feature method of [36]) makes online kernel learning approaches scalable in the sense that the learner can choose the number of parameters that should be learned, independent of the number of observed data samples (see e.g. [30, 4, 52]). Employing finite-dimensional feature representations of kernels to perform online federated kernel learning, clients can choose the number of parameters that they should send to the server. Therefore, finite-dimensional kernel approximation can better cope with limited clients-to-server communication bandwidth compared to conventional kernel learning approaches. Random feature (RF) approximation [36] has been exploited to perform online federated kernel learning when a single pre-selected kernel function is employed [26, 17]. The choice of the kernel function greatly affects the performance of function approximation when it comes to exploiting only a single kernel function. Employing multiple kernels instead of a single pre-selected one, can lead to obtaining more accurate function approximation since multi-kernel learning (MKL) can learn combination of kernels [24]. Online federated MKL algorithms with theoretical guarantees called vM-KOFL and eM-KOFL have been proposed in [22]. However, eM-KOFL and vM-KOFL do not provide personalized MKL models for clients since they learn the same combination of kernels for all clients.

The present paper proposes a novel **p**ersonalized **o**nline **f**ederated **MKL** algorithm called POF-MKL that provides a personalized MKL model for each client while it is ensured that the available clients-to-server communication bandwidth can afford communication cost of sending clients' updates to the server. In order to alleviate the communication cost of MKL, the propsoed POF-MKL employs RF approximation of kernels and at each time instant, each client chooses a subset of kernels to send their updates to the server instead of sending the updates of all kernels. The number of kernels in the chosen subset is selected such that the required bandwidth to send all clients' updates does not exceed the available clients-to-server communication bandwidth. Therefore, clients can send their updates to the server independent of the number of kernels in the dictionary and as a result a comparatively large dictionary of kernels can be considered to perform function approximation. Contributions of the present paper can be summarized as follows:

**c1.** Leveraging the proposed POF-MKL, clients can update a subset of kernels' parameters which alleviates computational complexity and communication cost of sending updates to the server;

**c2.** Through theoretical analysis, it is proved that using the proposed POF-MKL, each client achieves sub-linear regret with respect to RF approximation of the best kernel in hindsight associated with the corresponding client data samples (c.f. Theorem 1). Moreover, it is guaranteed that the server achieves sub-linear regret with respect to the best function approximator (c.f. Theorem 2);

**c3.** Experiments on real datasets showcase the effectiveness of the proposed POF-MKL compared to existing online federated kernel learning algorithms.

## 2   Problem Statement and Preliminaries

Let there be a set of $K$ clients performing function approximation task in an online fashion. The $k$-th client's goal is to learn the function $f$ using the stream of data samples $\{(\boldsymbol{x}_{k,t}, y_{k,t})\}_{t=1}^{T}$ such that $\boldsymbol{x}_{k,t} \in \mathbb{R}^d$ is the data sample observed by the $k$-th client at time $t$ and $y_{k,t}$ is the label associated with $\boldsymbol{x}_{k,t}$. In the kernel learning context, the function $f$ is assumed to belong to a reproducing kernel Hilbert space (RKHS). The present paper studies the personalized federated supervised function approximation problem

$$\min_{f \in \mathbb{H}} \sum_{t=1}^{T} \sum_{k=1}^{K} \mathcal{L}(f(\boldsymbol{x}_{k,t}), y_{k,t}) \tag{1}$$

where $\mathbb{H}$ represents the RKHS the function $f$ belongs to and $\mathcal{L}(\cdot, \cdot)$ denotes the loss function which can be defined as

$$\mathcal{L}(f(\boldsymbol{x}), y) = \mathcal{C}(f(\boldsymbol{x}), y) + \lambda \Omega(\|f\|^2) \tag{2}$$

where $\mathcal{C}(\cdot, \cdot)$ is the cost function (e.g. least squares function for regression task), $\lambda$ denotes the regularization coefficient and $\Omega(\cdot)$ represents a regularizer function to prevent over-fitting and control the model complexity. Let $\boldsymbol{\Theta}$ be the global parameters of the function $f$ which are learned through collaboration of clients with the server while $\boldsymbol{w}_k$ be the personalized parameter of the function $f$ learned locally by the $k$-th client. Thus, the goal is that the cumulative difference between $f(\boldsymbol{x}_{k,t}; \boldsymbol{\Theta}, \boldsymbol{w}_k)$ and $y_{k,t}$ over time is minimized. At each time instant $t$, upon observing the data sample $\boldsymbol{x}_{k,t}$, the $k$-th client make the prediction $f(\boldsymbol{x}_{k,t}; \boldsymbol{\Theta}, \boldsymbol{w}_k)$ and then observes the true label $y_{k,t}$. Therefore, the function approximation problem that the server aims at solving can be expressed as $\min_{\boldsymbol{\Theta}} \sum_{t=1}^{T} \sum_{k=1}^{K} \mathcal{L}(f(\boldsymbol{x}_{k,t}; \boldsymbol{\Theta}, \boldsymbol{w}_k), y_{k,t})$. Moreover, finding the local parameters $\boldsymbol{w}_k$ by the $k$-th client can be expressed as the optimization problem $\min_{\boldsymbol{w}_k} \sum_{t=1}^{T} \mathcal{L}(f(\boldsymbol{x}_{k,t}; \boldsymbol{\Theta}, \boldsymbol{w}_k), y_{k,t})$. In order to perform the function approximation task in an online fashion, the $k$-th client needs to perform the task with the values of $\boldsymbol{\Theta}$ and $\boldsymbol{w}_k$ at time $t$ denoted by $\boldsymbol{\Theta}_t$ and $\boldsymbol{w}_{k,t}$, respectively. Thus, the values of function parameters $\boldsymbol{\Theta}$ and $\boldsymbol{w}_k$ should be updated 'on the fly'. Since the function $f(\cdot; \cdot, \cdot)$ belongs to a reproducing kernel Hilbert space (RKHS), based on the representer theorem [48], given data samples, the optimal solution for (1) can be obtained as

$$\hat{f}(\boldsymbol{x}) = \sum_{t=1}^{T} \sum_{k=1}^{K} \alpha_{k,t} \kappa(\boldsymbol{x}, \boldsymbol{x}_{k,t}) \tag{3}$$

where $\kappa(\cdot, \cdot)$ denotes symmetric positive definite kernel function such that $\kappa(\boldsymbol{x}, \boldsymbol{x}')$ measures the similarity between $\boldsymbol{x}$ and $\boldsymbol{x}'$. And $\alpha_{k,t}$ is an unknown coefficient associated with $\kappa(\boldsymbol{x}, \boldsymbol{x}_{k,t})$ which is required to be estimated. In this case, $\hat{f}(\cdot)$ in (3) belongs to the RKHS $\mathbb{H} := \{f(\cdot) | f(\boldsymbol{x}) = \sum_{t=1}^{\infty} \sum_{k=1}^{K} \alpha_{k,t} \kappa(\boldsymbol{x}, \boldsymbol{x}_{k,t})\}$ such that RKHS norm is defined as $\|f\|_{\mathbb{H}}^2 := \sum_t \sum_{t'} \alpha_t \alpha_{t'} \kappa(\boldsymbol{x}_t, \boldsymbol{x}_{t'})$. Furthermore, from (3), it can be inferred that $\boldsymbol{\Theta}_t = [\alpha_{1,1}, \ldots, \alpha_{K,1}, \ldots, \alpha_{1,t}, \ldots, \alpha_{K,t}]$. Therefore, the number of coefficients $\{\alpha_{k,\tau}\}_{\tau=1}^t$, $\forall k$ that should be estimated to obtain $\hat{f}(\cdot)$ increases over time. This is known as *curse of dimensionality* [48] since the computational complexity of function approximation increases with time. This brings challenge for federated implementation of function approximation since dimension of updates that should be sent to the server by each client grows over time and when $T$ is large, the available communication bandwidth may not be enough for clients to send their updates.

In order to deal with the increasing number of unknown coefficients, one can employ random Fourier approximation [36]. Assume that $\kappa(\cdot)$ is a shift-invariant kernel meaning that $\kappa(\boldsymbol{x}, \boldsymbol{x}') = \kappa(\boldsymbol{x} - \boldsymbol{x}')$. Let $\pi_\kappa(\boldsymbol{\rho})$ denotes the Fourier transform of $\kappa(\cdot)$. If the kernel function $\kappa(\cdot)$ is normalized such that $\kappa(\boldsymbol{0}) = 1$, then $\pi_\kappa(\boldsymbol{\rho})$ can be viewed as a probability density function (PDF) (see e.g. [36]). Let $\boldsymbol{\rho}_1, \ldots, \boldsymbol{\rho}_D$ be a set of $D$ independent and identically distributed (i.i.d) vectors drawn from $\pi_\kappa(\cdot)$. Let the vector $\boldsymbol{z}(\boldsymbol{x})$ be defined as

$$\boldsymbol{z}(\boldsymbol{x}) = \frac{1}{\sqrt{D}} [\sin(\boldsymbol{\rho}_1^\top \boldsymbol{x}), \ldots, \sin(\boldsymbol{\rho}_D^\top \boldsymbol{x}), \cos(\boldsymbol{\rho}_1^\top \boldsymbol{x}), \ldots, \cos(\boldsymbol{\rho}_D^\top \boldsymbol{x})]. \tag{4}$$

Then, $\hat{\kappa}_r(\boldsymbol{x} - \boldsymbol{x}') = \boldsymbol{z}(\boldsymbol{x})^\top \boldsymbol{z}(\boldsymbol{x}')$ constitutes an unbiased estimator of $\kappa(\boldsymbol{x} - \boldsymbol{x}')$ and the random feature (RF) approximation of $\hat{f}(\boldsymbol{x})$ in (3) can be obtained as

$$\hat{f}_{\text{RF}}(\boldsymbol{x}) = \sum_{t=1}^{T} \sum_{k=1}^{K} \alpha_{k,t} \boldsymbol{z}(\boldsymbol{x}_{k,t})^\top \boldsymbol{z}(\boldsymbol{x}) := \boldsymbol{\theta}^\top \boldsymbol{z}(\boldsymbol{x}) \tag{5}$$

where in this case $\boldsymbol{\theta} = \sum_{t=1}^{T} \sum_{k=1}^{K} \alpha_{k,t} \boldsymbol{z}(\boldsymbol{x}_{k,t})$. According to (4), $\boldsymbol{z}(\boldsymbol{x}_{k,t})$ is a $2D$ vector and as a result it can be concluded that $\boldsymbol{\theta}$ is a $2D$ vector as well. Therefore, using RF approximation, the vector $\boldsymbol{\theta}$ should be estimated whose dimension does not grow over time.

The performance of a kernel learning algorithm depends on the choice of the kernel. Thus, performing the function approximation using a pre-selected kernel requires prior information which may not be available. To cope with this, employing a dictionary of kernels in lieu of a pre-selected single kernel has been proposed in the literature (see e.g. [45, 24, 31]). Specifically, the kernel is learned as a combination of kernels in the dictionary. Let $\kappa_1(\cdot), \ldots, \kappa_N(\cdot)$ be a set of

$N$ kernels where $\kappa_i(\cdot)$ denotes the $i$-th kernel. The function $\bar{\kappa}(\cdot)$ belongs to the convex hull $\mathbb{K} := \{\bar{\kappa}(\boldsymbol{x}) = \sum_{i=1}^{N} \beta_i \kappa_i(\boldsymbol{x}), \beta_i \geq 0, \forall i, \sum_{i=1}^{N} \beta_i = 1\}$ is a kernel [41]. Therefore, in online multi-kernel learning, the goal is to learn the convex combination of kernels in the dictionary to minimize the cumulative regret with respect to the best function approximator in hindsight. The cumulative regret is defined as the cumulative difference between loss of the online multi-kernel learning algorithm and that of the best function approximator in hindsight. Furthermore, for a dataset $\{(\boldsymbol{x}_t, y_t)\}_{t=1}^{T}$, the best function approximator is $f^*(\cdot) \in \arg\min_{f_i^*, i \in [N]} \sum_{t=1}^{T} \mathcal{L}(f_i^*(\boldsymbol{x}_t), y_t)$ where $f_i^*(\cdot) \in \arg\min_{f \in \mathbb{H}_i} \sum_{t=1}^{T} \mathcal{L}(f(\boldsymbol{x}_t), y_t)$ such that $\mathbb{H}_i$ is an RKHS induced by $\kappa_i(\cdot)$ and $[N] := \{1, \ldots, N\}$. Enabled by random feature approximation, centralized and scalable online multi-kernel learning algorithms have been proposed in literature (see e.g. [40, 43, 15]). The present paper proposes an algorithmic framework for personalized online federated MKL using RF approximation of kernels in the dictionary.

## 3 Personalized Online Federated Multi-Kernel Learning

The present section proposes an algorithmic framework for online federated multi-kernel learning which can deal with heterogeneous data among clients. To perform function approximation, RF approximations of kernel functions are employed. For the $i$-th kernel $\kappa_i$, vectors $\boldsymbol{\rho}_{i,1}, \ldots, \boldsymbol{\rho}_{i,D}$ are drawn i.i.d from $\pi_{\kappa_i}(\cdot)$ to construct the random feature vector $\boldsymbol{z}_i(\boldsymbol{x}) = \frac{1}{\sqrt{D}}[\sin(\boldsymbol{\rho}_{i,1}^\top \boldsymbol{x}), \ldots, \sin(\boldsymbol{\rho}_{i,D}^\top \boldsymbol{x}), \cos(\boldsymbol{\rho}_{i,1}^\top \boldsymbol{x}), \ldots, \cos(\boldsymbol{\rho}_{i,D}^\top \boldsymbol{x})]$. Then, at time instant $t$, the random feature approximation associated with $\kappa_i(\cdot)$ can be obtained as $\hat{f}_{\text{RF},it}(\boldsymbol{x}) = \boldsymbol{\theta}_{i,t}^\top \boldsymbol{z}_i(\boldsymbol{x})$ where $\boldsymbol{\theta}_{i,t}$ is the global function parameter associated the $i$-th kernel at time $t$.

### 3.1 Algorithm

At each time instant $t$, the server transmits global function parameters $\boldsymbol{\theta}_{i,t}, \forall i \in [N]$ to all clients. The $k$-th client, assigns the weight $w_{ik,t}$ to the $i$-th kernel which indicates the confidence of the $k$-th client at time $t$ in the function approximation given by the $i$-th kernel. Upon receiving new data sample $\boldsymbol{x}_{k,t}$, the $k$-th client performs the function approximation combining kernels' RF approximations as

$$\hat{f}(\boldsymbol{x}_{k,t}; \hat{\boldsymbol{\Theta}}_t, \boldsymbol{w}_{k,t}) = \sum_{i=1}^{N} \frac{w_{ik,t}}{W_{k,t}} \boldsymbol{\theta}_{i,t}^\top \boldsymbol{z}_i(\boldsymbol{x}_{k,t}) = \sum_{i=1}^{N} \frac{w_{ik,t}}{W_{k,t}} \hat{f}_{\text{RF},it}(\boldsymbol{x}_{k,t}; \boldsymbol{\theta}_{i,t}) \tag{6}$$

where $\hat{\boldsymbol{\Theta}}_t = [\boldsymbol{\theta}_{1,t}, \ldots, \boldsymbol{\theta}_{N,t}]$, $\boldsymbol{w}_{k,t} = [w_{1k,t}, \ldots, w_{Nk,t}]$ and $W_{k,t} = \sum_{i=1}^{N} w_{ik,t}$. As it can be inferred from (6), each client constructs its own personalized combination of kernels. Upon observing the true label $y_{k,t}$, the $k$-th client calculates the losses $\mathcal{L}(\hat{f}_{\text{RF},it}(\boldsymbol{x}_{k,t}; \boldsymbol{\theta}_{i,t}), y_{k,t}), \forall i \in [N]$. Then, the $k$-th client leverages calculated losses to locally update both global and local parameters. Let $\boldsymbol{\theta}_{ik,t+1}$ and $w_{ik,t+1}$ denote the $k$-th client's local updates of $\boldsymbol{\theta}_{i,t}$ and $w_{ik,t}$, respectively. Specifically, the $k$-th client utilizes multiplicative update rule to update $w_{ik,t}$ as

$$w_{ik,t+1} = w_{ik,t} \exp\left(-\eta_k \mathcal{L}(\hat{f}_{\text{RF},it}(\boldsymbol{x}_{k,t}; \boldsymbol{\theta}_{i,t}), y_{k,t})\right), \forall i \in [N] \tag{7}$$

where $\eta_k$ is the learning rate of the $k$-th client. Note that the $k$-th client ($\forall k \in [K]$) does not send its updated local parameter $\boldsymbol{w}_{k,t+1}$ to the server. Clients send their locally updated global parameters to the server (i.e. $\boldsymbol{\theta}_{ik,t+1}$). Aggregating local updates, the server updates global parameters to $\hat{\boldsymbol{\Theta}}_{t+1}$. If all clients send updates associated with all kernels (i.e. $\boldsymbol{\theta}_{ik,t+1}, \forall i \in [N]$), this requires sending $2ND$ parameters by each client at each time instant. When the number of both clients and kernels are large, the available client-to-server communication bandwidth may not be enough to afford sending $2NDK$ parameters per time instant even for small values of $D$. Note that reducing $N$ and $D$ degrade the performance of online federated MKL. Reducing $N$ (the number of kernels), decreases the flexibility of clients to construct their ideal kernel using convex combination of kernels in the dictionary. Reducing $D$ can degrade the accuracy of RF approximation.

The present paper proposes an algorithmic framework to enable clients to perform online function approximation with sufficiently large dictionary of kernels while the available clients-to-server communication bandwidth can afford sending updates from clients to the server when a desirable value for the number of random features $D$ is chosen. To this end, at each time instant, each client

---

**Algorithm 1** The $k$-th client kernel subset selection at time $t$.

---

**Input:** Weights $w_{ik,t}, \forall i \in [N]$, parameter $M$ and exploration rate $0 < \xi_k \leq 1$.

Sort the kernels in descending order with respect to weights $\{w_{ik,t}\}_{i=1}^{N}$.

Obtain the index sequence $s_1, \ldots, s_N$ such that $w_{s_b k,t} \leq w_{s_a k,t}$ if $b > a, \forall a, b \in [N]$.

Open bin $\mathcal{B}_1$ and initialize $j = 1$.

**for all** $i \in [N]$, the $k$-th client **do**

    **if** the bin $\mathcal{B}_j$ includes less than $M$ kernels **then**

        Adds the $s_i$-th kernel to $\mathcal{B}_j$.

    **else**

        Opens new bin $\mathcal{B}_{j+1}$, adds the $s_i$-th kernel to $\mathcal{B}_{j+1}$ and updates $j \leftarrow j + 1$.

    **end if**

**end for**

Draw an index $I_{k,t}$ via PMF $\boldsymbol{q}_{k,t}$ in (8).

**Output:** $\mathcal{S}_{k,t}$: indices set of kernels in the selected bin $\mathcal{B}_{I_{k,t}}$

---

randomly chooses a subset of $M$ kernels among all $N$ kernels in the dictionary. Then, each client updates and sends the global parameters of the chosen $M$ kernels to the server instead of updating and sending global parameters of all kernels. To choose a subset of $M$ kernels, each client splits kernels into some bins and draws randomly one of the bins at each time instant. Each bin contains at most $M$ kernels and each client updates and sends global parameters associated with kernels in the chosen bin. In order to distribute kernels among bins, at first the $k$-th client sorts kernels in descending order according to kernels' weights $\{w_{ik,t}\}_{i=1}^{N}$. Let $\mathcal{B}_j$ represents the $j$-th bin of kernels. The $k$-th client adds kernels from sorted list one by one to $\mathcal{B}_j$ until either all kernels are assigned to a bin or the number of kernels in $\mathcal{B}_j$ reaches $M$. When there are some kernels that are not assigned to any bins while there are $M$ kernels in $\mathcal{B}_j$, the $k$-th client opens the bin $\mathcal{B}_{j+1}$ and adds kernels to this bin. This continues until all kernels are assigned to a bin. As it can be inferred from the procedure of distributing kernels into bins, the number of bins at every client is $m = \left\lceil \frac{N}{M} \right\rceil$. Furthermore, it can be concluded that $\mathcal{B}_1$ includes $M$ kernels with the largest weights while the bin $\mathcal{B}_m$ includes $N - (m-1)M$ kernels with lowest weights. The $k$-th client assigns the weight $u_{jk,t}$ at time $t$ to $\mathcal{B}_j$ defined as $u_{jk,t} = \sum_{\kappa_i \in \mathcal{B}_j} w_{ik,t}$. The $k$-th client draws one of the bins according to the probability mass function (PMF) $\boldsymbol{q}_{k,t}$ defined as

$$q_{jk,t} = (1 - \xi_k)\frac{u_{jk,t}}{U_{k,t}} + \frac{\xi_k}{m}, \forall j \in [m] \tag{8}$$

where $U_{k,t} = \sum_{j=1}^{m} u_{jk,t}$ and $0 < \xi_k \leq 1$ is an exploration rate determined by the $k$-th client. Let $I_{k,t}$ be the index of the chosen bin by the $k$-th client at time $t$. The PMF in (8) constitutes trade-off between exploitation and exploration. According to the first term in the right hand side of (8), it is more probable that the $k$-th client draws a bin which includes kernels with larger weights $w_{ik,t}$. Hence, it is more probable that the $k$-th client collaborates in updating the global parameters of a kernel with larger weight $w_{ik,t}$. Let $\mathcal{S}_{k,t}$ denotes the set which includes the indices of kernels in the chosen bin at time $t$. The Algorithm 1 summarizes the procedure that the $k$-th client determines the set $\mathcal{S}_{k,t}$. According to Algorithm 1, kernel subset selection is personalized since each client chooses its own subset of kernels to update their parameters.

Let $p_{ik,t}$ denotes the probability that $i \in \mathcal{S}_{k,t}$. Then $p_{ik,t} = q_{b_i k,t}$ where $b_i$ is the index of the bin which includes the $i$-th kernel. The $k$-th client updates global parameters locally as follows

$$\boldsymbol{\theta}_{ik,t+1} = \boldsymbol{\theta}_{i,t} - \eta \frac{\nabla \mathcal{L}(\boldsymbol{\theta}_{i,t}^{\top} \boldsymbol{z}_i(\boldsymbol{x}_{k,t}), y_{k,t})}{p_{ik,t}} \mathbf{1}_{i \in \mathcal{S}_{k,t}} \tag{9}$$

where $\mathbf{1}_{i \in \mathcal{S}_{k,t}}$ denotes an indicator function and it is 1 when $i \in \mathcal{S}_{k,t}$. The update rule in (9) implies that when $i \notin \mathcal{S}_{k,t}$, the $k$-th client does not update $\boldsymbol{\theta}_{i,t}$ (i.e. $\boldsymbol{\theta}_{ik,t+1} = \boldsymbol{\theta}_{i,t}$). Therefore, the $k$-th client sends $\boldsymbol{\theta}_{ik,t+1}$ to the server only if $i \in \mathcal{S}_{k,t}$. Therefore, at each time instant, each client needs to send at most $2MD$ parameters to the server. Let $\mathcal{C}_{i,t}$ be a set of client indices such that $k \in \mathcal{C}_{i,t}$ if the $k$-th client sends $\boldsymbol{\theta}_{ik,t+1}$ to the server. Upon aggregating updates from clients, the server updates $\boldsymbol{\theta}_{i,t}$ as

$$\boldsymbol{\theta}_{i,t+1} = \boldsymbol{\theta}_{i,t} - \frac{1}{K} \sum_{k \in \mathcal{C}_{i,t}} (\boldsymbol{\theta}_{i,t} - \boldsymbol{\theta}_{ik,t+1}) = \boldsymbol{\theta}_{i,t} - \frac{\eta}{K} \sum_{k=1}^{K} \frac{\nabla \mathcal{L}(\boldsymbol{\theta}_{i,t}^{\top} \boldsymbol{z}_i(\boldsymbol{x}_{k,t}), y_{k,t})}{p_{ik,t}} \mathbf{1}_{i \in \mathcal{S}_{k,t}}. \tag{10}$$

---

**Algorithm 2** Personalized Online Federated Multi-Kernel Learning (POF-MKL)

---

**Input:** Kernels $\kappa_i$, $i = 1, ..., N$, learning rate $\eta > 0$ and the number of random features $D$.
**Initialize:** $\boldsymbol{\theta}_{i,1} = \mathbf{0}$, $w_{ik,1} = 1$, $\forall i \in [N], \forall k \in [K]$.
**for** $t = 1, \ldots, T$ **do**
    The server transmits the global parameters $\hat{\boldsymbol{\Theta}}_t = [\boldsymbol{\theta}_{1,t}, \ldots, \boldsymbol{\theta}_{N,t}]$ to all clients.
    **for all** $k \in [K]$, the $k$th client **do**
        Receive one datum $\boldsymbol{x}_{k,t}$.
        Predicts $\hat{f}(\boldsymbol{x}_{k,t}; \hat{\boldsymbol{\Theta}}_t, \boldsymbol{w}_{k,t})$ via (6).
        Calculates losses $\mathcal{L}(\hat{f}_{\mathrm{RF},it}(\boldsymbol{x}_{k,t}; \boldsymbol{\theta}_{i,t}), y_{k,t})$, $\forall i \in [N]$.
        Updates $w_{ik,t+1}$, $\forall i \in [N]$ via (7).
        Selects a subset of kernel indices $\mathcal{S}_{k,t}$ using Algorithm 1.
        Updates $\boldsymbol{\theta}_{ik,t+1}$, $\forall i \in \mathcal{S}_{k,t}$ via (9) and sends $\boldsymbol{\theta}_{ik,t+1}$, $\forall i \in \mathcal{S}_{k,t}$ to the server.
    **end for**
    The server updates $\boldsymbol{\theta}_{i,t+1}$, $\forall i \in [N]$ via (10).
**end for**

---

Algorithm 2 summarizes the proposed personalized online federated multi-kernel learning algorithm called POF-MKL. It is useful to note that using our proposed POF-MKL, the server cannot find the gradients $\nabla \mathcal{L}(\boldsymbol{\theta}_{i,t}^\top \boldsymbol{z}_i(\boldsymbol{x}_{k,t}), y_{k,t})$ from updates received from clients. Instead, the server can find $\nabla \mathcal{L}(\boldsymbol{\theta}_{i,t}^\top \boldsymbol{z}_i(\boldsymbol{x}_{k,t}), y_{k,t})/p_{ik,t}$ where $p_{ik,t}$ is a time-varying value determined locally by the $k$-th client. This can promote the privacy of the proposed POF-MKL since exchanging the gradients can be hazardous to the privacy of federated learning (see e.g. [53, 14]).

**Complexity.** Each client needs to store $d$-dimensional $D$ random feature vectors for each kernel. Therefore, the memory requirement of each client to implement function approximation using POF-MKL is $\mathcal{O}(dND)$. Using POF-MKL, at each time instant, each client needs to perform $\mathcal{O}(dND)$ operations including inner products and summations. Furthermore, when $\xi_k < 1$, in order to choose a subset of kernels, the $k$-th client needs to sort kernels which imposes worst case computational complexity of $\mathcal{O}(N \log N)$. However, when $\xi_k = 1$, according to PMF in (8), the $k$-th client chooses one bin uniformly at random and as a result in this case the $k$-th client does not need to sort kernels. Therefore, setting $\xi_k < 1$, the computational complexity for the $k$-th client is $\mathcal{O}(dND + N \log N)$ while setting $\xi_k = 1$, the computational complexity of the $k$-th client at each time instant is $\mathcal{O}(dND)$.

### 3.2 Regret Analysis

The present section analyzes the regret of the proposed POF-MKL. Specifically, two types of regret $\mathcal{R}_{k,T}$ and $\mathcal{R}_{s,T}$ are considered for the $k$-th client and the server, respectively. The performance of the $k$-th client utilizing POF-MKL is analyzed in terms of regret defined as

$$\mathcal{R}_{k,T} = \sum_{t=1}^{T} \mathcal{L}(\hat{f}(\boldsymbol{x}_{k,t}; \hat{\boldsymbol{\Theta}}_t, \boldsymbol{w}_{k,t}), y_{k,t}) - \min_{i \in [N]} \sum_{t=1}^{T} \mathcal{L}(\hat{f}_{\mathrm{RF},it}(\boldsymbol{x}_{k,t}; \boldsymbol{\theta}_{i,t}), y_{k,t}) \tag{11}$$

where $\mathcal{R}_{k,T}$ measures the cumulative difference between the loss of the $k$-th client and the loss of the RF approximation of the kernel with minimum loss among all kernels' RF approximations. Let $\alpha_{ik,t}^*$, $\forall t \in [T], \forall k \in [K]$ represents the optimal coefficients associated with the $i$-th kernel such that $f_i^*(\boldsymbol{x}) = \sum_{t=1}^{T} \sum_{k=1}^{K} \alpha_{ik,t}^* \kappa_i(\boldsymbol{x}, \boldsymbol{x}_{k,t})$. Then the best function approximator is defined as $f^*(\cdot) \in \arg\min_{f_i^*, i \in [N]} \sum_{t=1}^{T} \sum_{k=1}^{K} \mathcal{L}(f_i^*(\boldsymbol{x}_{k,t}), y_{k,t})$. Furthermore, the regret of the server is defined as the cumulative difference between the loss of POF-MKL and that of the best function approximator over all data samples distributed among clients which can be expressed as

$$\mathcal{R}_{s,T} = \sum_{t=1}^{T} \sum_{k=1}^{K} \mathcal{L}(\hat{f}(\boldsymbol{x}_{k,t}; \hat{\boldsymbol{\Theta}}_t, \boldsymbol{w}_{k,t}), y_{k,t}) - \sum_{t=1}^{T} \sum_{k=1}^{K} \mathcal{L}(f^*(\boldsymbol{x}_{k,t}), y_{k,t}). \tag{12}$$

In order to analyze the regret of POF-MKL, suppose that the following assumptions hold true:

**(as1)** $\mathcal{L}(\boldsymbol{\theta}_{i,t}^\top \boldsymbol{z}_i(\boldsymbol{x}_{k,t}), y_{k,t})$, $\forall k \in [K]$ is convex with respect to $\boldsymbol{\theta}_{i,t}$ at each time instant $t$.
**(as2)** For $\boldsymbol{\theta}$ in a bounded set satisfying $\|\boldsymbol{\theta}\| \leq C$, the loss function and its gradient are bounded

as $0 \leq \mathcal{L}(\boldsymbol{\theta}^\top \boldsymbol{z}_i(\boldsymbol{x}_{k,t}), y_{k,t}) \leq 1$ and $\|\nabla \mathcal{L}(\boldsymbol{\theta}^\top \boldsymbol{z}_i(\boldsymbol{x}_{k,t}), y_{k,t})\| \leq L$. Moreover, each data sample is bounded as $\|\boldsymbol{x}_{k,t}\| \leq 1, \forall k \in [K], \forall t \in [T]$.

**(as3)** Kernels $\kappa_i(\cdot), \forall i \in [N]$ are shift-invariant with $\kappa_i(\boldsymbol{0}) = 1, \forall i \in [N]$.

The following theorem investigates the regret of the $k$-th client according to the $k$-th client data. The proof of the following Theorem can be found in Appendix A.

**Theorem 1.** *Under (as1)–(as3), the regret of the $k$-th client with respect to the best kernel satisfies*

$$\sum_{t=1}^{T} \mathcal{L}(\hat{f}(\boldsymbol{x}_{k,t}; \hat{\boldsymbol{\Theta}}_t, \boldsymbol{w}_{k,t}), y_{k,t}) - \min_{i \in [N]} \sum_{t=1}^{T} \mathcal{L}(\hat{f}_{RF,it}(\boldsymbol{x}_{k,t}; \boldsymbol{\theta}_{i,t}), y_{k,t}) \leq \frac{\ln N}{\eta_k} + \frac{\eta_k}{2} T. \quad (13)$$

Theorem 1 shows that by setting $\eta_k = \mathcal{O}\left(\frac{1}{\sqrt{T}}\right)$, the $k$-th client achieves sub-linear regret of $\mathcal{O}(\sqrt{T})$. Furthermore, Theorem 1 shows that POF-MKL can deal with heterogeneous data among clients since the regret of each client defined in (11) is calculated with respect to the corresponding client data. The following theorem studies the regret of the server with respect to the best function approximator. The proof can be found in Appendix B.

**Theorem 2.** *Let $i^* := \arg\min_{i \in [N]} \sum_{t=1}^{T} \sum_{k=1}^{K} \mathcal{L}(f_i^*(\boldsymbol{x}_{k,t}), y_{k,t})$ and $\sigma_i$ be the second Fourier moment of the $i$-th kernel. Under (as1)–(as3), the regret of the server with respect to the best function approximator satisfies*

$$\sum_{t=1}^{T} \sum_{k=1}^{K} \mathcal{L}(\hat{f}(\boldsymbol{x}_{k,t}; \hat{\boldsymbol{\Theta}}_t, \boldsymbol{w}_{k,t}), y_{k,t}) - \sum_{t=1}^{T} \sum_{k=1}^{K} \mathcal{L}(f^*(\boldsymbol{x}_{k,t}), y_{k,t})$$

$$\leq \frac{KC^2}{2\eta} + \frac{\eta}{2} \sum_{t=1}^{T} \sum_{k=1}^{K} \frac{L^2}{p_{i^*k,t}} + \sum_{k=1}^{K} \left( \frac{\ln N}{\eta_k} + \frac{\eta_k}{2} T \right) + \epsilon LKTC \quad (14)$$

*with probability at least $1 - 2^8 \left(\frac{\sigma_{i^*}}{\epsilon}\right)^2 \exp\left(-\frac{D\epsilon^2}{4(d+2)}\right)$ where $C := \max_{i \in [N]} \sum_{t=1}^{T} \sum_{k=1}^{K} \alpha_{ik,t}^*$.*

As it can be inferred from (14), the regret of the server with respect to the best function approximator depends on $\frac{1}{p_{i^*k,t}}$. From (8) and the fact that $p_{ik,t} = q_{b_i k,t}$, it can be concluded that $p_{i^*k,t} > \frac{\xi_k}{m}$. Thus, setting $\xi_k = \mathcal{O}(1)$, then $p_{ik,t} > \mathcal{O}(\frac{M}{N})$. The regret bound in (14) shows that setting $\eta = \mathcal{O}\left(\sqrt{\frac{M}{NT}}\right)$ and $\epsilon = \eta_k = \frac{1}{\sqrt{T}}, \forall k \in [K]$, the server obtains regret of $\mathcal{O}\left(\sqrt{\frac{N}{M}T}\right)$ with probability at least $1 - 2^8 \left(\frac{\sigma_{i^*}}{\epsilon}\right)^2 \exp\left(-\frac{D\epsilon^2}{4(d+2)}\right)$. This shows that increasing $M$ tighten the regret bound and increasing $D$ increases the probability that the regret bound in (14) holds true. However, using POF-MKL, each client needs to transmit $MD$ parameters at each time instant. Since both $M$ and $D$ are determined by the algorithm POF-MKL, this shows that POF-MKL can provide flexibility to tighten regret bound while the available clients-to-server communication bandwidth can afford transmission of clients' updates to the server. It is useful to mention that choosing larger value for $\xi_k$ increases the lower bound of $p_{i^*k,t}$ and as a result the optimal choice for $\xi_k$ in terms of regret is $\xi_k = 1$. However, choosing smaller values for $\xi_k$ makes the value of $p_{ik,t}$ more dependent on weights $\{w_{ik,t}\}_{k=1}^{K}$ (c.f. (8)). Therefore, choosing smaller values for $\xi_k$ makes $p_{ik,t}$ less predictable. This makes estimating $\nabla \mathcal{L}(\boldsymbol{\theta}_{i,t}^\top z_i(x_{k,t}), y_{k,t})$ given $\nabla \mathcal{L}(\boldsymbol{\theta}_{i,t}^\top z_i(x_{k,t}), y_{k,t})/p_{ik,t}$ more difficult which leads to better protection of privacy.

**Comparison with personalized federated learning.** In order to deal with heterogeneous data among clients, personalized federated learning has been studied extensively in the literature (see [44, 8, 9, 12, 20, 10, 29, 28, 42, 6, 1, 32, 50, 2, 46, 5]). Utilizing model-agnostic meta-learning [13], personalized federated learning algorithms have been proposed in [12, 1]. In [42, 5], personalized federated learning algorithms have been designed by learning hyper-networks [18]. In [32], a personalized model is a linear combination of a set of shared component models such that each client constructs its personalized mixture of models. However, in aforementioned personalized federated learning works, clients are assumed to store a dataset to perform local updates with. Therefore, when clients are not able to store data in batch and they have to make a decision upon receiving a new data sample, aforementioned works in personalized federated learning cannot guarantee sub-linear

regret for clients. However, according to Theorems 1 and 2, POF-MKL provides sub-linear regret for clients when clients cannot store data in batch and make decision in an online fashion.

**Comparison with online federated learning [34].** Fed-OMD algorithm has been proposed in [34] which enables clients to perform their learning task in an online and federated fashion while it is proved that Fed-OMD enjoys sub-linear regret when the loss function is convex with respect to parameters required to be learnt at each time instant. The proposed POF-MKL differs from Fed-OMD in the sense that Fed-OMD cannot guarantee sub-linear regret when it comes to performing the online learning task with RF approximations of multiple kernels since the loss function $\mathcal{L}(\sum_{i=1}^{N} w_{i,t}\boldsymbol{\theta}_{i,t}^{\top}\boldsymbol{z}_i(\boldsymbol{x}), y)$ is not convex with respect to both $\boldsymbol{\theta}_{i,t}$ and $w_{i,t}$. However, according to Theorems 1 and 2, the proposed POF-MKL guarantees sub-linear regret.

**Comparison with [22].** Online federated MKL algorithms called vM-KOFL and eM-KOFL have been presented in [22]. Both POF-MKL and algorithms in [22] exploit random feature approximation to alleviate computational complexity of online kernel learning. Furthermore, both POF-MKL and algorithms in [22] learn a linear combination of kernels. The proposed POF-MKL has the following advantages and innovations compared to vM-KOFL and eM-KOFL: i) The proposed POF-MKL allows clients to learn their own personalized combination of kernels (c.f. (6)). As it is proved in Theorem 1, the proposed POF-MKL can deal with heterogeneous data among clients in the sense that using POF-MKL each client guarantees sub-linear regret with respect to the best kernel RF approximation according to the corresponding client data. However, both vM-KOFL and eM-KOFL are not able to provide such guarantee. ii) Using vM-KOFL, each client needs to send $(N+1)D$ parameters to the server. However, using the proposed POF-MKL, each client needs to send $MD$ parameters to the server such that $M \leq N$ is determined by POF-MKL and can be chosen to be much smaller than $N$. iii) In eM-KOFL, the server chooses a kernel at each time instant and clients send their local updates associated with the chosen kernel by the server. The proposed POF-MKL provides more flexibility compared to eM-KOFL in the sense that using POF-MKL each client can send local updates of $M \geq 1$ kernels to the server. And each client chooses its own subset of kernels to send their updates to the server. Therefore, even though POF-MKL sets $M$ to 1, it is possible that at a time instant the server receives updates associated with all kernels in the dictionary. It is useful to mention that using eM-KOFL the cumulative regret of all clients is sub-linear with respect to the best kernel RF approximation with probability $1 - \delta$ where $0 < \delta \leq 1$. However, utilizing the update rule in (9), using the proposed POF-MKL, each client obtains sub-linear regret with respect to RF approximation of its best kernel with probability 1.

## 4 Experiments

We tested the performance of the proposed POF-MKL for online regression task through a set of experiments. The performance of POF-MKL is compared with the baselines PerFedAvg [12], OFSKL [34], OFMKL-Avg [34], vM-KOFL [22] and eM-KOFL [22]. PerFedAvg refers to the personalized federated averaging algorithm in [12]. In the experiments, PerFedAvg employs a fully connected feedforward neural network model. More information about the implementation of PerFedAvg can be found in Appendix C. OFSKL and OFMKL-Avg are two variations of Fed-OMD [34]. OFSKL leverages Fed-OMD [34] when a single radial basis function (RBF) with bandwidth of 10 is employed to perform the learning task. In OFMKL-Avg, kernels are learned independently from each other using Fed-OMD [34] and the prediction is the average of approximations given by kernels. Moreover, vM-KOFL and eM-KOFL are online federated MKL algorithms of [22] such that vM-KOFL requires transmission of all kernel updates at every time instant while eM-KOFL requires transmission of a kernel update at each time instant. In the experiments, each client observes 500 samples until the end of the learning task meaning that $T = 500$. The performance of the proposed POF-MKL and other baselines are tested on the following real datasets downloaded from UCI machine learning repository [11]: Naval [7], UJI [47], Air [51] and WEC [35]. More detailed information about datasets can be found in Appendix C. Data samples of Naval and UJI datasets are distributed i.i.d among clients. Data samples in Air and WEC datasets are distributed non-i.i.d among clients. More inforamtion about distributing data samples among clients can be found in Appendix C. The number of clients for Naval, UJI, Air and WEC datasets are 23, 42, 240 and 560, respectively. The dictionary of kernels consists of 51 RBFs with different bandwidth such that the bandwidth of the $i$-th kernel is $\sigma_i = 10^{\frac{2i-52}{25}}$. We consider the case where the clients-to-server communication bandwidth is limited such that at each time instant, the maximum number of parameters that a client is allowed to transmit to the server

is 1000. Furthermore, the memory and computational capability of clients are limited such that the maximum value can be picked for the number of random features $D$ is 100. The experiments were carried for 20 different sets of random feature vectors. The performance of algorithms is measured using average of mean squared error (MSE) defined as

$$\text{MSE} = \frac{1}{20} \sum_{j=1}^{20} \frac{1}{KT} \sum_{t=1}^{T} \sum_{k=1}^{K} (\hat{y}_{jk,t} - y_{k,t})^2$$

where $\hat{y}_{jk,t}$ denotes the prediction of the $k$-th client at time instant $t$ corresponding to the $j$-th set of random feature vectors. Learning rates are set to $\eta = \eta_k = \frac{1}{\sqrt{T}}, \forall k$. Also, exploration rates are set to $\xi_k = 1, \forall k$. The performance of POF-MKL with different $\xi_k$ is studied in Appendix C. Codes are available at `https://github.com/pouyamghari/POF-MKL`.

Table 1 presents the MSE and run time performance of online federated kernel learning algorithms on real datasets. Run time refers to average total run time of clients to perform online learning task on the entire data samples that they observe. In Table 1, $M$ refers to the number of kernels whose updates are sent to the server after prediction at each time instant. And, $D$ is the number of random features. Comparing MSE of POF-MKL with that of OFMKL-Avg, it can be concluded that learning the weights to combine kernels provides higher accuracy than averaging kernels' predictions. Table 1 shows that POF-MKL with $M = 1$ provides lower MSE than eM-KOFL. Using eM-KOFL, at each time instant, the server receives updates belong to only one kernel. However, using POF-MKL with $M = 1$, each client sends an update belongs to a kernel which is selected by the client. Therefore, the server receives updates associated with different kernels even though $M = 1$. Therefore, experimental results show the effectiveness of the personalized kernel selection provided by POF-MKL. It can be observed that POF-MKL with $M = 25$ obtains lower MSE than those of POF-MKL with $M = 51$ and vM-KOFL. Since each client is allowed to send at most 1000 parameters per time instant, if clients send updates of all kernels at every time instant as this is the case in vM-KOFL, $D$ cannot be chosen to be greater than 9. However, setting $M = 25$, POF-MKL can set $D = 20$ which can improve the accuracy of online regression task compared to the case where $D = 9$. Note that according to Theorem 2, increase in $D$ increases the probability that the server achieves sub-linear regret with respect to the best function approximator. Furthermore, POF-MKL with $M = 51$ achieves lower MSE than vM-KOFL even if data samples are distributed i.i.d among clients. This shows that the proposed POF-MKL can better cope with heterogeneous data among clients which is in agreement with theoretical results in Theorem 1. In fact, the optimal combination of kernels can be different across clients. Using POF-MKL, each client constructs its own personalized combination of kernels which results in lower MSE compared to vM-KOFL. The proposed POF-MKL with $M = 1$ and $M = 25$ runs faster than eM-KOFL. In fact, using POF-MKL, clients only need to update parameters associated with $M$ kernels while employing vM-KOFL and eM-KOFL, clients have to update parameters of all kernels. Moreover, POF-MKL obtains lower MSE than PerFedAvg. Note that since clients are not able to store data in batch, at each time instant clients update PerFedAvg's model using only the newly observed data sample. Therefore, convergence of PerFedAvg is not guaranteed. Experimental results show that POF-MKL achieves higher accuracy than PerFedAvg in online regression task when it is not possible for clients to store data in batch. Since OFSKL employs only a pre-selected single kernel, OFSKL runs faster than POF-MKL. However, utilizing multiple kernels enables POF-MKL to obtain lower MSE than that of OFSKL. In fact, using POF-MKL clients learn a linear combination of kernels which is proved to enjoy sub-linear regret with respect to the best kernel in hindsight while employing OFSKL clients have to make predictions using a pre-selected kernel. Furthermore, Figure 1 illustrates the average regret of clients when clients employ vM-KOFL and the proposed POF-MKL with different $M$ parameters. From Figure 1, it can be observed that the proposed POF-MKL achieves sub-linear regret.

## 5 Conclusions

The present paper proposed a personalized online federated MKL algorithm called POF-MKL based on RF approximation. Employing the proposed POF-MKL, each client updates the parameters of a subset of kernels which alleviates the computational complexity of the client as well as communication cost of sending updated parameters of kernels. Theoretical analysis proved that using POF-MKL, each client achieves sub-linear regret with respect to the RF approximation of its best kernel in hindsight which indicates that POF-MKL can deal heterogeneous data among clients. While each

Table 1: MSE($\times 10^{-3}$) and run time of online federated learning algorithms on real datasets.

| Algorithms | $M$ | $D$ | MSE($\times 10^{-3}$) | | | | Run time(s) | | | |
|---|---|---|---|---|---|---|---|---|---|---|
| | | | Naval | UJI | Air | WEC | Naval | UJI | Air | WEC |
| PerFedAvg | - | - | 118.60 | 63.03 | 13.68 | 77.33 | 44.59 | 41.67 | 37.40 | 33.56 |
| OFSKL | 1 | 100 | 77.77 | 61.82 | 13.65 | 87.87 | 0.07 | 0.06 | 0.08 | 0.06 |
| OFMKL-Avg | 51 | 9 | 33.25 | 55.44 | 10.63 | 34.01 | 1.51 | 1.73 | 0.91 | 0.47 |
| vM-KOFL | 51 | 9 | 26.42 | 51.50 | 10.58 | 25.17 | 2.01 | 2.22 | 1.37 | 0.67 |
| eM-KOFL | 1 | 100 | 28.64 | 61.08 | 21.94 | 20.14 | 2.27 | 10.13 | 1.45 | 1.70 |
| POF-MKL | 1 | 100 | **16.16** | **33.02** | **9.27** | **11.44** | 1.22 | 9.02 | 1.25 | 1.10 |
| POF-MKL | 25 | 20 | 16.82 | 37.34 | 9.34 | 11.58 | 0.69 | 2.29 | 0.63 | 0.52 |
| POF-MKL | 51 | 9 | 16.65 | 41.00 | 9.38 | 11.97 | 0.82 | 1.07 | 0.81 | 0.65 |

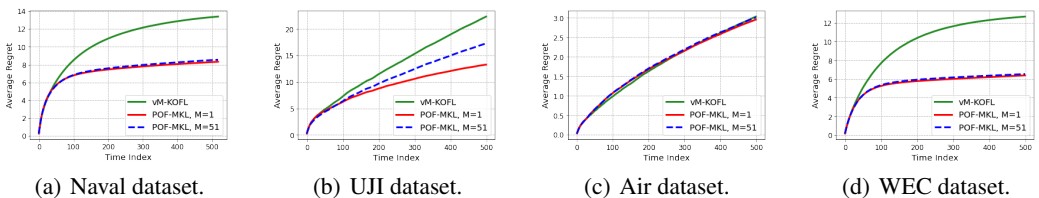

(a) Naval dataset.  (b) UJI dataset.  (c) Air dataset.  (d) WEC dataset.

Figure 1: Average regret of clients.

client updates a subset of kernels, it was proved that the server achieves sub-linear regret with respect to the best function approximator. Experiments on real datasets showcased the advantages of POF-MKL compared with other online federated kernel learning algorithms.

## Acknowledgement

This work is supported by NSF ECCS 2207457. PI Yanning Shen also receives support from Microsoft Academic Grant Award for AI Research. Contact: Yanning Shen (yannings@uci.edu).

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
