Using the inequality $e^{-x} \le 1 - x + \frac{1}{2}x^2, \forall x \ge 0$, the upper bound of (15) can be obtained as

$$\frac{W_{k,t+1}}{W_{k,t}} \le \sum_{i=1}^{N} \frac{w_{ik,t}}{W_{k,t}} \left(1 - \eta_k \mathcal{L}(\hat{f}_{\text{RF},it}(\boldsymbol{x}_{k,t}; \boldsymbol{\theta}_{i,t}), y_{k,t}) + \frac{\eta_k^2}{2} \mathcal{L}^2(\hat{f}_{\text{RF},it}(\boldsymbol{x}_{k,t}; \boldsymbol{\theta}_{i,t}), y_{k,t})\right). \tag{16}$$

Using the inequality $1 + x \le e^x$ and taking the logarithm from both sides of (16), we get

$$\ln \frac{W_{k,t+1}}{W_{k,t}} \le \sum_{i=1}^{N} \frac{w_{ik,t}}{W_{k,t}} \left(-\eta_k \mathcal{L}(\hat{f}_{\text{RF},it}(\boldsymbol{x}_{k,t}; \boldsymbol{\theta}_{i,t}), y_{k,t}) + \frac{\eta_k^2}{2} \mathcal{L}^2(\hat{f}_{\text{RF},it}(\boldsymbol{x}_{k,t}; \boldsymbol{\theta}_{i,t}), y_{k,t})\right). \tag{17}$$

According to (as2), $\mathcal{L}^2(\hat{f}_{\text{RF},it}(\boldsymbol{x}_{k,t}; \boldsymbol{\theta}_{i,t}), y_{k,t}) \le 1$. Therefore, from (17), we can conclude that

$$\ln \frac{W_{k,t+1}}{W_{k,t}} \le \sum_{i=1}^{N} \frac{w_{ik,t}}{W_{k,t}} \left(\frac{\eta_k^2}{2} - \eta_k \mathcal{L}(\hat{f}_{\text{RF},it}(\boldsymbol{x}_{k,t}; \boldsymbol{\theta}_{i,t}), y_{k,t})\right). \tag{18}$$

Summing (18) over time, we arrive at

$$\ln \frac{W_{k,T+1}}{W_{k,1}} \le \sum_{t=1}^{T} \sum_{i=1}^{N} \frac{w_{ik,t}}{W_{k,t}} \left(\frac{\eta_k^2}{2} - \eta_k \mathcal{L}(\hat{f}_{\text{RF},it}(\boldsymbol{x}_{k,t}; \boldsymbol{\theta}_{i,t}), y_{k,t})\right). \tag{19}$$

Moreover, for any $i \in [N]$, $\ln \frac{W_{k,T+1}}{W_{k,1}}$ can be lower bounded as

$$\ln \frac{W_{k,T+1}}{W_{k,1}} \ge \ln \frac{w_{ik,T+1}}{W_{k,1}} = -\eta_k \sum_{t=1}^{T} \mathcal{L}(\hat{f}_{\text{RF},it}(\boldsymbol{x}_{k,t}; \boldsymbol{\theta}_{i,t}), y_{k,t}) - \ln N. \tag{20}$$

Combining (19) with (20), we obtain

$$\sum_{t=1}^{T} \sum_{i=1}^{N} \frac{w_{ik,t}}{W_{k,t}} \mathcal{L}(\hat{f}_{\text{RF},it}(\boldsymbol{x}_{k,t}; \boldsymbol{\theta}_{i,t}), y_{k,t}) - \sum_{t=1}^{T} \mathcal{L}(\hat{f}_{\text{RF},it}(\boldsymbol{x}_{k,t}; \boldsymbol{\theta}_{i,t}), y_{k,t}) \le \frac{\ln N}{\eta_k} + \frac{\eta_k}{2}T. \tag{21}$$

Since the loss function $\mathcal{L}(\cdot, \cdot)$ is convex, using (21) and Jensen inequality we can write

$$\sum_{t=1}^{T} \mathcal{L}\left(\sum_{i=1}^{N} \frac{w_{ik,t}}{W_{k,t}} \hat{f}_{\text{RF},it}(\boldsymbol{x}_{k,t}; \boldsymbol{\theta}_{i,t}), y_{k,t}\right) - \sum_{t=1}^{T} \mathcal{L}(\hat{f}_{\text{RF},it}(\boldsymbol{x}_{k,t}; \boldsymbol{\theta}_{i,t}), y_{k,t}) \le \frac{\ln N}{\eta_k} + \frac{\eta_k}{2}T \tag{22}$$

which proves the Theorem 1.

# B  Proof of Theorem 2

In order to prove Theorem 2, the following Lemma is used as a stepstone.

**Lemma 3.** *Let $\alpha_{ik,t}^*$, $\forall t \in [T]$, $\forall k \in [K]$ represents the optimal coefficients associated with the $i$-th kernel such that $f_i^*(\boldsymbol{x}) = \sum_{t=1}^{T} \sum_{k=1}^{K} \alpha_{ik,t}^* \kappa_i(\boldsymbol{x}, \boldsymbol{x}_{k,t})$. And $\hat{f}_i^*(\boldsymbol{x}) = (\boldsymbol{\theta}_i^*)^\top \boldsymbol{z}_i(\boldsymbol{x})$ denotes the best RF-based estimator associated with the $i$-th kernel such that $\boldsymbol{\theta}_i^* = \sum_{t=1}^{T} \sum_{k=1}^{K} \alpha_{ik,t}^* \boldsymbol{z}_i(\boldsymbol{x}_{k,t})$. Under assumptions (as1)–(as3), using POF-MKL, the RF approximation of the $i$-th kernel satisfies*

$$\sum_{t=1}^{T} \sum_{k=1}^{K} \mathcal{L}(\hat{f}_{RF,it}(\boldsymbol{x}_{k,t}; \boldsymbol{\theta}_{i,t}), y_{k,t}) - \sum_{t=1}^{T} \sum_{k=1}^{K} \mathcal{L}(\hat{f}_i^*(\boldsymbol{x}_{k,t}), y_{k,t})$$

$$\le \frac{K \|\boldsymbol{\theta}_i^*\|^2}{2\eta} + \frac{\eta}{2} \sum_{t=1}^{T} \sum_{k=1}^{K} \frac{L^2}{p_{ik,t}}. \tag{23}$$

*Proof.* Let $\ell_{ik,t}$ be the importance sampling loss estimate defined as

$$\ell_{ik,t} = \frac{\mathcal{L}(\boldsymbol{\theta}_{i,t}^\top \boldsymbol{z}_i(\boldsymbol{x}_{k,t}), y_{k,t})}{p_{ik,t}} \mathbf{1}_{i \in \mathcal{S}_{k,t}}. \tag{24}$$

Then according to (9), for any fixed $\boldsymbol{\theta}$, it can be written that

$$\left\| \frac{1}{K} \sum_{k=1}^K \boldsymbol{\theta}_{ik,t+1} - \boldsymbol{\theta} \right\|^2 = \left\| \frac{1}{K} \sum_{k=1}^K (\boldsymbol{\theta}_{i,t} - \eta \nabla \ell_{ik,t}) - \boldsymbol{\theta} \right\|^2 = \left\| \boldsymbol{\theta}_{i,t} - \boldsymbol{\theta} - \frac{\eta}{K} \sum_{k=1}^K \nabla \ell_{ik,t} \right\|^2$$

$$= \|\boldsymbol{\theta}_{i,t} - \boldsymbol{\theta}\|^2 - \frac{2\eta}{K} \left( \sum_{k=1}^K \nabla^\top \ell_{ik,t} \right) (\boldsymbol{\theta}_{i,t} - \boldsymbol{\theta}) + \left\| \frac{\eta}{K} \sum_{k=1}^K \nabla \ell_{ik,t} \right\|^2 \tag{25}$$

According to the convexity of the loss function $\mathcal{L}(\boldsymbol{\theta}^\top \boldsymbol{x}, y)$ with respect to $\boldsymbol{\theta}$ as stated in (as1), we find

$$\mathcal{L}(\boldsymbol{\theta}_{i,t}^\top \boldsymbol{z}_i(\boldsymbol{x}_{k,t}), y_{k,t}) - \mathcal{L}(\boldsymbol{\theta}^\top \boldsymbol{z}_i(\boldsymbol{x}_{k,t}), y_{k,t}) \leq \nabla^\top \mathcal{L}(\boldsymbol{\theta}_{i,t}^\top \boldsymbol{z}_i(\boldsymbol{x}_{k,t}), y_{k,t})(\boldsymbol{\theta}_{i,t} - \boldsymbol{\theta}). \tag{26}$$

Multiplying both sides of (26) by $\frac{\mathbf{1}_{i \in \mathcal{S}_{k,t}}}{p_{ik,t}}$, we get

$$\left( \frac{\mathcal{L}(\boldsymbol{\theta}_{i,t}^\top \boldsymbol{z}_i(\boldsymbol{x}_{k,t}), y_{k,t})}{p_{ik,t}} - \frac{\mathcal{L}(\boldsymbol{\theta}^\top \boldsymbol{z}_i(\boldsymbol{x}_{k,t}), y_{k,t})}{p_{ik,t}} \right) \mathbf{1}_{i \in \mathcal{S}_{k,t}}$$

$$\leq \frac{\nabla^\top \mathcal{L}(\boldsymbol{\theta}_{i,t}^\top \boldsymbol{z}_i(\boldsymbol{x}_{k,t}), y_{k,t})}{p_{ik,t}} (\boldsymbol{\theta}_{i,t} - \boldsymbol{\theta}) \mathbf{1}_{i \in \mathcal{S}_{k,t}}. \tag{27}$$

Summing (27) over $k$, $\forall k \in [K]$, we arrive at

$$\sum_{k=1}^K \left( \frac{\mathcal{L}(\boldsymbol{\theta}_{i,t}^\top \boldsymbol{z}_i(\boldsymbol{x}_{k,t}), y_{k,t})}{p_{ik,t}} - \frac{\mathcal{L}(\boldsymbol{\theta}^\top \boldsymbol{z}_i(\boldsymbol{x}_{k,t}), y_{k,t})}{p_{ik,t}} \right) \mathbf{1}_{i \in \mathcal{S}_{k,t}}$$

$$\leq \left( \sum_{k=1}^K \frac{\nabla^\top \mathcal{L}(\boldsymbol{\theta}_{i,t}^\top \boldsymbol{z}_i(\boldsymbol{x}_{k,t}), y_{k,t})}{p_{ik,t}} \mathbf{1}_{i \in \mathcal{S}_{k,t}} \right) (\boldsymbol{\theta}_{i,t} - \boldsymbol{\theta}). \tag{28}$$

Based on the definition of $\ell_{ik,t}$, (28) can be rewritten as

$$\sum_{k=1}^K \ell_{ik,t} - \sum_{k=1}^K \frac{\mathcal{L}(\boldsymbol{\theta}^\top \boldsymbol{z}_i(\boldsymbol{x}_{k,t}), y_{k,t})}{p_{ik,t}} \mathbf{1}_{i \in \mathcal{S}_{k,t}} \leq \left( \sum_{k=1}^K \nabla^\top \ell_{ik,t} \right) (\boldsymbol{\theta}_{i,t} - \boldsymbol{\theta}). \tag{29}$$

According to (25), (29) is equivalent to

$$\sum_{k=1}^K \ell_{ik,t} - \sum_{k=1}^K \frac{\mathcal{L}(\boldsymbol{\theta}^\top \boldsymbol{z}_i(\boldsymbol{x}_{k,t}), y_{k,t})}{p_{ik,t}} \mathbf{1}_{i \in \mathcal{S}_{k,t}}$$

$$\leq \frac{K}{2\eta} \left( \|\boldsymbol{\theta}_{i,t} - \boldsymbol{\theta}\|^2 - \left\| \frac{1}{K} \sum_{k=1}^K \boldsymbol{\theta}_{ik,t+1} - \boldsymbol{\theta} \right\|^2 + \left\| \frac{\eta}{K} \sum_{k=1}^K \nabla \ell_{ik,t} \right\|^2 \right) \tag{30}$$

Expectations of $\ell_{ik,t}$ and $\|\nabla \ell_{ik,t}\|^2$ with respect to $\mathbf{1}_{i \in \mathcal{S}_{k,t}}$ can be calculated as

$$\mathbb{E}_t[\ell_{ik,t}] = \frac{\mathcal{L}(\boldsymbol{\theta}_{i,t}^\top \boldsymbol{z}_i(\boldsymbol{x}_{k,t}), y_{k,t})}{p_{ik,t}} p_{ik,t} = \mathcal{L}(\boldsymbol{\theta}_{i,t}^\top \boldsymbol{z}_i(\boldsymbol{x}_{k,t}), y_{k,t}) \tag{31a}$$

$$\mathbb{E}_t[\|\nabla \ell_{ik,t}\|^2] = \frac{\|\nabla \mathcal{L}(\boldsymbol{\theta}_{i,t}^\top \boldsymbol{z}_i(\boldsymbol{x}_{k,t}), y_{k,t})\|^2}{p_{ik,t}^2} p_{ik,t} = \frac{\|\nabla \mathcal{L}(\boldsymbol{\theta}_{i,t}^\top \boldsymbol{z}_i(\boldsymbol{x}_{k,t}), y_{k,t})\|^2}{p_{ik,t}}. \tag{31b}$$

Furthermore, using AM-GM inequality and (31b), it can be concluded that

$$\mathbb{E}_t \left[ \| \sum_{k=1}^K \nabla \ell_{ik,t} \|^2 \right] \leq \mathbb{E}_t \left[ K \sum_{k=1}^K \|\nabla \ell_{ik,t}\|^2 \right] \leq K \sum_{k=1}^K \frac{\|\nabla \mathcal{L}(\boldsymbol{\theta}_{i,t}^\top \boldsymbol{z}_i(\boldsymbol{x}_{k,t}), y_{k,t})\|^2}{p_{ik,t}}. \tag{32}$$

According to (10), considering the fact that

$$\|\boldsymbol{\theta}_{i,t+1} - \boldsymbol{\theta}\|^2 = \left\|\frac{1}{K}\sum_{k=1}^{K}\boldsymbol{\theta}_{ik,t+1} - \boldsymbol{\theta}\right\|^2$$

taking the expectation from (30) with respect to $\mathbf{1}_{i\in\mathcal{S}_{k,t}}, \forall k \in [K]$ leads to

$$\sum_{k=1}^{K}\mathcal{L}(\boldsymbol{\theta}_{i,t}^{\top}\boldsymbol{z}_i(\boldsymbol{x}_{k,t}), y_{k,t}) - \sum_{k=1}^{K}\mathcal{L}(\boldsymbol{\theta}^{\top}\boldsymbol{z}_i(\boldsymbol{x}_{k,t}), y_{k,t})$$

$$\leq \frac{K}{2\eta}\left(\|\boldsymbol{\theta}_{i,t} - \boldsymbol{\theta}\|^2 - \|\boldsymbol{\theta}_{i,t+1} - \boldsymbol{\theta}\|^2\right) + \frac{\eta}{2}\sum_{k=1}^{K}\frac{\|\nabla\mathcal{L}(\boldsymbol{\theta}_{i,t}^{\top}\boldsymbol{z}_i(\boldsymbol{x}_{k,t}), y_{k,t})\|^2}{p_{ik,t}}. \tag{33}$$

According to (as2), we can conclude that $\|\nabla\mathcal{L}(\boldsymbol{\theta}_{i,t}^{\top}\boldsymbol{z}_i(\boldsymbol{x}_{k,t}), y_{k,t})\|^2 \leq L^2$. Hence, summing (33) over time, given the fact that $\boldsymbol{\theta}_{i,1} = \boldsymbol{0}, \forall i \in [N]$, we get

$$\sum_{t=1}^{T}\sum_{k=1}^{K}\mathcal{L}(\boldsymbol{\theta}_{i,t}^{\top}\boldsymbol{z}_i(\boldsymbol{x}_{k,t}), y_{k,t}) - \sum_{t=1}^{T}\sum_{k=1}^{K}\mathcal{L}(\boldsymbol{\theta}^{\top}\boldsymbol{z}_i(\boldsymbol{x}_{k,t}), y_{k,t})$$

$$\leq \frac{K}{2\eta}\left(\|\boldsymbol{\theta}\|^2 - \|\boldsymbol{\theta}_{i,T+1} - \boldsymbol{\theta}\|^2\right) + \frac{\eta}{2}\sum_{t=1}^{T}\sum_{k=1}^{K}\frac{L^2}{p_{ik,t}}. \tag{34}$$

Replacing $\boldsymbol{\theta}$ with $\boldsymbol{\theta}_i^*$ and considering the fact that $\|\boldsymbol{\theta}_{i,T+1} - \boldsymbol{\theta}\|^2 \geq 0$, we obtain

$$\sum_{t=1}^{T}\sum_{k=1}^{K}\mathcal{L}(\boldsymbol{\theta}_{i,t}^{\top}\boldsymbol{z}_i(\boldsymbol{x}_{k,t}), y_{k,t}) - \sum_{t=1}^{T}\sum_{k=1}^{K}\mathcal{L}((\boldsymbol{\theta}_i^*)^{\top}\boldsymbol{z}_i(\boldsymbol{x}_{k,t}), y_{k,t}) \leq \frac{K\|\boldsymbol{\theta}_i^*\|^2}{2\eta} + \frac{\eta}{2}\sum_{t=1}^{T}\sum_{k=1}^{K}\frac{L^2}{p_{ik,t}}$$

which proves the Lemma 3. $\qquad\square$

In order to proof Theorem 2, we leverage the results obtained in the proofs of Lemma 3 and Theorem 1. Since (22) holds true for any $i$, summing (22) over all $k \in [K]$, for any $i$ we can write

$$\sum_{t=1}^{T}\sum_{k=1}^{K}\mathcal{L}(\hat{f}(\boldsymbol{x}_{k,t};\hat{\boldsymbol{\Theta}}_t, \boldsymbol{w}_{k,t}), y_{k,t}) - \sum_{t=1}^{T}\sum_{k=1}^{K}\mathcal{L}(\hat{f}_{\text{RF},it}(\boldsymbol{x}_{k,t};\boldsymbol{\theta}_{i,t}), y_{k,t})$$

$$\leq \sum_{k=1}^{K}\left(\frac{\ln N}{\eta_k} + \frac{\eta_k}{2}T\right). \tag{35}$$

Combining (35) with (23), we arrive at

$$\sum_{t=1}^{T}\sum_{k=1}^{K}\mathcal{L}(\hat{f}(\boldsymbol{x}_{k,t};\hat{\boldsymbol{\Theta}}_t, \boldsymbol{w}_{k,t}), y_{k,t}) - \sum_{t=1}^{T}\sum_{k=1}^{K}\mathcal{L}(\hat{f}_i^*(\boldsymbol{x}_{k,t}), y_{k,t})$$

$$\leq \frac{K\|\boldsymbol{\theta}_i^*\|^2}{2\eta} + \frac{\eta}{2}\sum_{t=1}^{T}\sum_{k=1}^{K}\frac{L^2}{p_{ik,t}} + \sum_{k=1}^{K}\left(\frac{\ln N}{\eta_k} + \frac{\eta_k}{2}T\right). \tag{36}$$

According to claim 1 in [36], it can be written that $\sup_{\boldsymbol{x},\boldsymbol{x}'}|\boldsymbol{z}_i^{\top}(\boldsymbol{x})\boldsymbol{z}_i(\boldsymbol{x}') - \kappa_i(\boldsymbol{x},\boldsymbol{x}')| \leq \epsilon$ holds true with probability greater than $1 - 2^8\left(\frac{\sigma_i}{\epsilon}\right)^2\exp\left(-\frac{D\epsilon^2}{4(d+2)}\right)$ where $\sigma_i$ is the second Fourier moment of the $i$-th kernel $\kappa_i(\cdot)$. Furthermore, according to (as3), the loss function is $L$-Lipschitz continuous and as a result it can be written that

$$\sum_{t=1}^{T}\sum_{k=1}^{K}|\mathcal{L}(\hat{f}_i^*(\boldsymbol{x}_{k,t}), y_{k,t}) - \mathcal{L}(f_i^*(\boldsymbol{x}_{k,t}), y_{k,t})|$$

$$\leq \sum_{t=1}^{T}\sum_{k=1}^{K}L\left|\sum_{\tau=1}^{T}\sum_{j=1}^{K}\alpha_{ij,\tau}^*\boldsymbol{z}_i^{\top}(\boldsymbol{x}_{j,\tau})\boldsymbol{z}_i(\boldsymbol{x}_{k,t}) - \sum_{\tau=1}^{T}\sum_{j=1}^{K}\alpha_{ij,\tau}^*\kappa_i(\boldsymbol{x}_{j,\tau}, \boldsymbol{x}_{k,t})\right|. \tag{37}$$

Applying Cauchy-Schwartz inequality to the right hand side of (37), the left hand side of (37) can be bounded from above as

$$\sum_{t=1}^{T}\sum_{k=1}^{K}|\mathcal{L}(\hat{f}_i^*(\boldsymbol{x}_{k,t}), y_{k,t}) - \mathcal{L}(f_i^*(\boldsymbol{x}_{k,t}), y_{k,t})|$$

$$\leq \sum_{t=1}^{T}\sum_{k=1}^{K} L \sum_{\tau=1}^{T}\sum_{j=1}^{K} |\alpha_{ij,\tau}^*||\boldsymbol{z}_i^\top(\boldsymbol{x}_{j,\tau})\boldsymbol{z}_i(\boldsymbol{x}_{k,t}) - \kappa_i(\boldsymbol{x}_{j,\tau}, \boldsymbol{x}_{k,t})|. \tag{38}$$

Let $C := \max_{i \in [N]} \sum_{t=1}^{T}\sum_{k=1}^{K} \alpha_{ik,t}^*$. Therefore, we can conclude that

$$\sum_{t=1}^{T}\sum_{k=1}^{K}|\mathcal{L}(\hat{f}_i^*(\boldsymbol{x}_{k,t}), y_{k,t}) - \mathcal{L}(f_i^*(\boldsymbol{x}_{k,t}), y_{k,t})| \leq \epsilon LKTC \tag{39}$$

with probability at least $1 - 2^8 \left(\frac{\sigma_i}{\epsilon}\right)^2 \exp\left(-\frac{D\epsilon^2}{4(d+2)}\right)$. Moreover, using Triangle inequality, we can write

$$\sum_{t=1}^{T}\sum_{k=1}^{K}\mathcal{L}(\hat{f}_i^*(\boldsymbol{x}_{k,t}), y_{k,t}) - \sum_{t=1}^{T}\sum_{k=1}^{K}\mathcal{L}(f_i^*(\boldsymbol{x}_{k,t}), y_{k,t})$$

$$\leq \left|\sum_{t=1}^{T}\sum_{k=1}^{K}\mathcal{L}(\hat{f}_i^*(\boldsymbol{x}_{k,t}), y_{k,t}) - \mathcal{L}(f_i^*(\boldsymbol{x}_{k,t}), y_{k,t})\right|$$

$$\leq \sum_{t=1}^{T}\sum_{k=1}^{K}|\mathcal{L}(\hat{f}_i^*(\boldsymbol{x}_{k,t}), y_{k,t}) - \mathcal{L}(f_i^*(\boldsymbol{x}_{k,t}), y_{k,t})| \leq \epsilon LKTC \tag{40}$$

which holds true with probability at least $1 - 2^8 \left(\frac{\sigma_i}{\epsilon}\right)^2 \exp\left(-\frac{D\epsilon^2}{4(d+2)}\right)$. Moreover, for $\boldsymbol{z}_i^\top(\boldsymbol{x})\boldsymbol{z}_i(\boldsymbol{x}')$ we can write

$$\boldsymbol{z}_i^\top(\boldsymbol{x})\boldsymbol{z}_i(\boldsymbol{x}') = \frac{1}{D}\sum_{j=1}^{D}(\sin(\boldsymbol{\rho}_{i,j}^\top\boldsymbol{x})\sin(\boldsymbol{\rho}_{i,j}^\top\boldsymbol{x}') + \cos(\boldsymbol{\rho}_{i,j}^\top\boldsymbol{x})\cos(\boldsymbol{\rho}_{i,j}^\top\boldsymbol{x}')). \tag{41}$$

Based on arithmetic-mean geometric-mean, (41) can be relaxed to

$$\boldsymbol{z}_i^\top(\boldsymbol{x})\boldsymbol{z}_i(\boldsymbol{x}') \leq \frac{1}{D}\sum_{j=1}^{D}\frac{1}{2}(\sin^2(\boldsymbol{\rho}_{i,j}^\top\boldsymbol{x}) + \sin^2(\boldsymbol{\rho}_{i,j}^\top\boldsymbol{x}') + \cos^2(\boldsymbol{\rho}_{i,j}^\top\boldsymbol{x}) + \cos^2(\boldsymbol{\rho}_{i,j}^\top\boldsymbol{x}')) = 1. \tag{42}$$

Thus, given the fact that $|\boldsymbol{z}_i^\top(\boldsymbol{x})\boldsymbol{z}_i(\boldsymbol{x}')| \leq 1$, $\|\boldsymbol{\theta}_i^*\|^2$ can be bounded as

$$\|\boldsymbol{\theta}_i^*\|^2 \leq \sum_{t=1}^{T}\sum_{k=1}^{K}\sum_{\tau=1}^{T}\sum_{j=1}^{K}|\alpha_{ik,t}^*\alpha_{ij,\tau}^*\boldsymbol{z}_i^\top(\boldsymbol{x}_{j,\tau})\boldsymbol{z}_i(\boldsymbol{x}_{k,t})| \leq C^2. \tag{43}$$

Combining (40) and (43) with (36) yields

$$\sum_{t=1}^{T}\sum_{k=1}^{K}\mathcal{L}(\hat{f}(\boldsymbol{x}_{k,t}; \hat{\boldsymbol{\Theta}}_t, \boldsymbol{w}_{k,t}), y_{k,t}) - \sum_{t=1}^{T}\sum_{k=1}^{K}\mathcal{L}(f_i^*(\boldsymbol{x}_{k,t}), y_{k,t})$$

$$\leq \frac{KC^2}{2\eta} + \frac{\eta}{2}\sum_{t=1}^{T}\sum_{k=1}^{K}\frac{L^2}{p_{ik,t}} + \sum_{k=1}^{K}\left(\frac{\ln N}{\eta_k} + \frac{\eta_k}{2}T\right) + \epsilon LKTC \tag{44}$$

which holds true for any $i \in [N]$ with probability at least $1 - 2^8 \left(\frac{\sigma_i}{\epsilon}\right)^2 \exp\left(-\frac{D\epsilon^2}{4(d+2)}\right)$. Therefore, this proves the Theorem 2.

Table 2: MSE($\times 10^{-3}$) and standard deviation($\times 10^{-3}$) of online federated learning algorithms on real datasets.

| Algorithms | $M$ | $D$ | Naval | UJI | Air | WEC |
|---|---|---|---|---|---|---|
| OFSKL | 1 | 100 | $77.77 \pm 1.04$ | $61.82 \pm 2.76$ | $13.65 \pm 0.61$ | $87.87 \pm 3.93$ |
| OFMKL-Avg | 51 | 9 | $33.25 \pm 1.46$ | $55.44 \pm 2.48$ | $10.63 \pm 0.47$ | $34.01 \pm 1.52$ |
| vM-KOFL | 51 | 9 | $26.42 \pm 1.16$ | $51.50 \pm 2.30$ | $10.58 \pm 0.47$ | $25.17 \pm 1.12$ |
| eM-KOFL | 1 | 100 | $28.64 \pm 1.32$ | $61.08 \pm 2.73$ | $21.94 \pm 1.16$ | $20.14 \pm 0.93$ |
| POF-MKL | 1 | 100 | $\mathbf{16.16 \pm 0.72}$ | $\mathbf{33.02 \pm 1.48}$ | $\mathbf{9.27 \pm 0.41}$ | $\mathbf{11.44 \pm 0.52}$ |
| POF-MKL | 25 | 20 | $16.82 \pm 0.74$ | $37.34 \pm 1.67$ | $9.34 \pm 0.42$ | $11.58 \pm 0.53$ |
| POF-MKL | 51 | 9 | $16.65 \pm 0.74$ | $41.00 \pm 1.83$ | $9.38 \pm 0.42$ | $11.97 \pm 0.55$ |

## C  Supplementary Experimental Results and Details

This section presents further experimental results testing different aspects of the proposed algorithm POF-MKL. Moreover, this section provides more detailed information about experimental setup associated with results in section 4. The performance of federated kernel learning algorithms are tested on the following datasets:

- **Naval:** The dataset consists of $11,500$ samples. Each sample has 15 features of a a naval vessel. The goal is to predict lever position [7].

- **UJI:** The dataset consists of $21,000$ data samples. Each data sample has $520$ features which are WiFi fingerprints. The goal is to predict the geographical longitude associated with each data sample.

- **Air:** The dataset consists of $120,000$ samples with 14 features including information related to air quality such as concentration of some chemicals in the air. Data samples are collected from 4 different geographical sites. The goal is to predict the concentration of CO in the air [51]. For each site there are $30,000$ samples in the dataset.

- **WEC:** The dataset consists of $280,000$ samples with $48$ features of wave energy converters. Data samples are collected from 4 different geographical sites. The goal is to predict total power output [35]. For each site, there are $70,000$ samples.

Data samples of Naval and UJI datasets are distributed i.i.d among clients. The number of clients for Naval and UJI datasets are 23 and 42, respectively. Data samples in Air and WEC datasets are distributed non-i.i.d among clients. The number of clients for Air and WEC datasets are 240 and 560, respectively. For both Air and WEC datasets, there are 4 different geographical sites that each sample belongs to one of them. Each client observes 350 samples from one site and 50 samples from each of the rest of 3 sites. Moreover, PerFedAvg uses a feedforward neural network model. Each layer is a fully-connected dense layer with at most 20 neurons. Neurons in hidden layers exploit ReLU activation functions. Since each client cannot transmit more than 1000 parameters to the server, the number of hidden layers is determined in a way that the number of the neural network's parameters to be less than 1000. The number of parameters depends on the number of features in data samples. Therefore, the number of hidden layers varies across different datasets. For each dataset, given the number of features, the maximum number of hidden layers with 20 neurons is chosen. All experiments were carried out using Intel(R) Core(TM) i7-10510U CPU @ 1.80 GHz 2.30 GHz processor with a 64-bit Windows operating system.

Table 2 presents average MSE along with MSE standard deviation calculated over 20 different sets of random feature vectors. As it can be seen from Table 2, the proposed POF-MKL provides lower standard deviation compared to all other baselines. This shows that the proposed POF-MKL is less sensitive to the choice of random features. Furthermore, Table 3 reports the average cumulative regret of clients along with the standard deviation of regret among clients. As it can be inferred from Table 3, the proposed POF-MKL obtains lower regret than other online federated MKL algorithms. Moreover, for Air and WEC datasets, the standard deviation of regret among clients associated with POF-MKL is considerably lower than those of other online federated MKL algorithms. Note that data samples in Air and WEC datasets are distributed non-i.i.d among clients. Therefore, the results in Table 3 confirm that the proposed POF-MKL can better deal with heterogeneous data among clients.

Table 3: Average regret and its standard deviation across clients for online federated MKL learning algorithms.

| Algorithms | $M$ | $D$ | Naval | UJI | Air | WEC |
|---|---|---|---|---|---|---|
| OFMKL-Avg | 51 | 9 | $16.95 \pm 0.39$ | $24.23 \pm 20.33$ | $3.05 \pm 2.83$ | $17.07 \pm 14.52$ |
| vM-KOFL | 51 | 9 | $13.40 \pm 0.39$ | $22.28 \pm 15.56$ | $3.02 \pm 2.79$ | $12.65 \pm 11.53$ |
| eM-KOFL | 1 | 100 | $14.92 \pm 0.65$ | $27.26 \pm 19.79$ | $8.53 \pm 2.51$ | $10.43 \pm 8.28$ |
| POF-MKL | 1 | 100 | $\mathbf{8.33 \pm 0.39}$ | $\mathbf{13.23 \pm 8.80}$ | $\mathbf{2.95 \pm 2.08}$ | $\mathbf{6.37 \pm 5.28}$ |
| POF-MKL | 25 | 20 | $8.67 \pm 0.39$ | $15.41 \pm 9.58$ | $2.98 \pm 2.12$ | $6.41 \pm 5.39$ |
| POF-MKL | 51 | 9 | $8.55 \pm 0.39$ | $17.23 \pm 10.07$ | $3.01 \pm 2.15$ | $6.51 \pm 5.57$ |

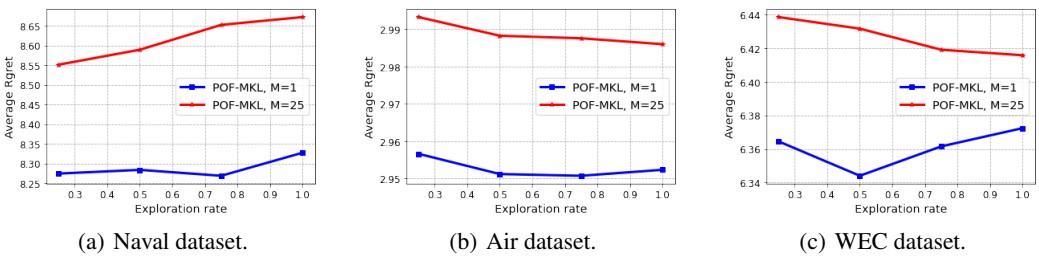

(a) Naval dataset.    (b) Air dataset.    (c) WEC dataset.

Figure 2: Average cumulative regret of clients with the change in the value of exploration rate ($\xi_k$).

Figure 2 illustrates the average regret of clients employing POF-MKL with the change in the value of exploration rate $\xi_k$ when the exploration rate of all clients are the same. In particular, Figure 2 depicts the performance of POF-MKL for $M = 1$ and $M = 25$ with the change in $\xi_k$. According to the PMF $\boldsymbol{q}_{k,t}$ defined in (8), the increase in $\xi_k$ leads to increase in exploration such that if $\xi_k = 1$, the $k$-th client chooses a subset of kernels uniformly at random. Figure 2 indicates that the optimal choice of $\xi_k$ in terms of regret depends on the dataset distributed among clients as well as the number of chosen kernels $M$. Moreover, the choice of $\xi_k$ is related to the computational complexity of executing POF-MKL by clients. Specifically, when $\xi_k < 1$, in order to choose a subset of kernels, the $k$-th client needs to sort kernels which imposes worst case computational complexity of $\mathcal{O}(N \log N)$. However, when $\xi_k = 1$, according to PMF in (8), the $k$-th client chooses one bin uniformly at random and as a result in this case the $k$-th client does not need to sort kernels. Also, it is useful to note that as it can be inferred from (9), clients can leverage the exploration rate $\xi_k$ to send their updates to the server without revealing both the gradient of loss and the loss of kernels which can promote the privacy of the proposed POF-MKL.