# OpenReview forum: "Personalized Online Federated Learning with Multiple Kernels"
_NeurIPS.cc/2022/Conference — NeurIPS 2022 Accept_

### Official Review · Reviewer_dLL3 · 2022-07-04

**Rating:** 4
**Confidence:** 4
**Soundness:** 3 good
**Presentation:** 3 good
**Contribution:** 3 good

**Summary:**

Current federated multi-kernel learning algorithms face challenges from high communication efficiency and heterogeneous data. This paper proposes a personalized online federated multi-kernel learning to overcome these challenges. Random feature approximation is used to improve the communication efficiency and the kernel subset selection is applied to achieve personalization. The theoretical analysis gives the upper bound of the regret for both the individual clients and the server. Numerical simulations are provided to show the effectiveness of the proposed algorithm.

**Questions:**

- Please explain how to choose the dictionary of kernels for different tasks.
- Does the regret have a lower bound? If so, can it show the optimal of the proposed method?
- What are the advantages and disadvantages of multiple-kernel learning compared with deep learning? Please explain why the paper uses multiple-kernel learning to achieve personalized online federated learning.


**Limitations:**

Yes

**Strengths And Weaknesses:**

Strengths
- The paper applies the random feature approximation and the kernel subset selection to achieve a personalized online federated multi-kernel learning algorithm. And numerical simulations show the advantages over other federated multi-kernel learning algorithms.
- The theoretical analysis is solid and supports their proposed algorithm well.

Weaknesses
- Random feature approximation was used in the online federated multi-kernel learning algorithm. The core contribution is that the paper achieves personalization via kernel subset selection. One may think the contribution is not very large.
- The literature review and comparisons are not enough. Since the paper focuses on personalized federated learning, the authors should cite and compare their algorithm with other personalized federated learning papers in the absence of multi-kennel learning.
- Experiments for other datasets should be added. In federated learning, it’s usual to compare different algorithms in datasets like MNIST, FMNIST, CIFAR-100, and even IMAGENET. The current datasets have small features.

---

> ### Author Response · Authors · 2022-08-02
> **Response to comments**
>
> We would like to thank the reviewer for the insightful comments. We have addressed the comments and accordingly modified the manuscript. Changes in the revised manuscript are highlighted in red color. Please find below our responses to comments:
>
> * *Contribution*: Personalized kernel subset selection brings challenges in regret analysis. In order to achieve sub-linear regret while clients perform personalized kernel selection, the update rule in equation (9) is proposed which is significantly different from update rules of prior online federated kernel learning algorithms. We hope that the reviewer takes this innovation into account to evaluate the novelty of this work.
> * *Literature review*: We added a discussion about personalized federated learning and cited more papers. Also, we compare previous works in personalized federated learning with the proposed POF-MKL in the added discussion. This revision can be found in lines 262-273 of the revised manuscript and is highlighted as red. We explained that previous works on personalized federated learning belong to offline cases and their theoretical guarantees cannot be applied to online learning tasks which is the main focus of this paper. In addition, previous personalized federated learning approaches require that clients store data samples in batch for training. However, the present paper assumes that clients may not be able to store observed data samples in batch. Using the proposed POF-MKL, clients do not need to store data samples.
> * *Comparison with personalized federated learning*: Furthermore, we have added PerFedAvg (personalized federated averaging of reference [12] in the revised version) to the baselines. The results can be seen in Table 1 of the revised manuscript. PerFedAvg uses a feedforward neural network model. Note that since clients are not able to store data in batch, in the experiments at each time instant, clients update PerFedAvg's model using only the newly observed data sample. Therefore, convergence of PerFedAvg is not guaranteed. Experimental results show that POF-MKL obtains lower MSE than PerFedAvg in online regression tasks when it is not possible for clients to store data in batch.
> * *Experiments for other datasets*: Kernel learning methods are less popular for image classification tasks since kernel learning methods would consider features in each pixel independent of other pixels. Instead, they are widely used to perform regression tasks (see e.g. [21], [38] and [39] of the revised manuscript). Therefore, in the experiments, we tested the performance of algorithms for online regression tasks. Also, to address the concern of small feature size, we have added a new dataset UJI whose samples have 520 features. The results can be observed in Table 1 of the revised manuscript. Results in Table 1 show that on the dataset UJI, POF-MKL obtains lower MSE than other alternative methods.

---

> ### Author Response · Authors · 2022-08-02
> **Response to questions**
>
> Please find below our responses to your questions:
>
> * *How to choose the dictionary of kernels*: Choosing the appropriate dictionary requires prior information about the performance of kernels on the learning task. When there is no prior information, one thing that can be done is to include a large number of kernels. This point brings us to one of the novelties of the present paper. Using the proposed POF-MKL, clients can employ a large number of kernels for their learning tasks while they can update a subset of kernels. When the number of kernels is large, the clients-to-server communication bandwidth may not be enough to send the updated parameters for all kernels.
> * *Regret lower bound*: The regret of the proposed algorithm does not have a lower bound, which is the same as other existing online federated learning literature.
> * *Deep learning vs kernel learning for online federated learning*: Kernel learning has the following advantages over deep learning when it comes to performing online learning tasks:
>
>     - Employing online multi-learning approaches to perform online learning tasks provides sublinear regret with respect to the best function approximator (the best kernel with the best parameters). However, employing neural networks for online learning tasks, there are no such theoretical guarantees for the performance. Moreover, in some cases for example due to shortage of memory, clients may not be able to store observed data samples in batch. Using online multi-learning approaches, clients do not need to store data samples in batch to obtain sub-linear regret. Furthermore, it is well-known that neural networks suffer from catastrophic forgetting, meaning that neural networks tend to forget previously learned information upon receiving new information (see e.g. [R1]). In online learning tasks where it is not possible to store data samples in a batch, neural networks may be overfitted to newly observed data samples. Therefore, when it comes to performing online learning tasks, kernel learning methods are more reliable than deep learning.
>
>      - Usually random feature based kernel learning methods are much more computationally efficient than deep neural networks. This makes kernel learning methods suitable for online federated learning tasks where clients have to make decisions quickly before the start of the next round. Thus, kernel learning might be a better option than deep learning when clients with limited computational capability (e.g. inexpensive sensor nodes in an internet of things network) want to perform online federated learning tasks. Experimental results in Table 1 of the revised manuscript also show that the proposed POF-MKL runs much faster than PerFedAvg which employs a feedforward neural network. Updating neural networks using backpropagation requires more computations than updating the parameters of random feature approximations of kernels.
>
> **References**
>
> [R1] V. V. Ramasesh, A. Lewkowycz, E. Dyer, “Effect of scale on catastrophic forgetting in neural networks,” International Conference on Learning Representations, 2022.

---

### Official Review · Reviewer_wHGi · 2022-07-11

**Rating:** 6
**Confidence:** 4
**Soundness:** 4 excellent
**Presentation:** 3 good
**Contribution:** 3 good

**Summary:**

This paper proposed a novel personalized online federated MKL framework (named POF-MKL) based on RF approximate. In particular, each client updates a subset of kernel parameters to alleviate computational complexity and achieve communication efficiency. Theoretical analysis shows the proposed method achieves sub-linear regret.

The authors empirically evaluated the proposed method on several datasets and it seems that the proposed method exhibited impressive performance and outperformed the baseline methods.

The main contributions of the paper include:

- Proposing the personalized federated MKL method to tackle the heterogeneous in federated learning.
- Proving the convergence rate of the proposed algorithm.
- Saving communication cost compared to existing algorithms, both theoretically and experimentally.

**Questions:**

1. I am curious to know how to consider the case when only a subset of clients participated at every iteration in online federation learning. Would it have an impact on convergence ?
2. I would like to know how equation 7 is derived.

**Limitations:**

N.A.

**Strengths And Weaknesses:**

Prons:

1. The paper is technically sound and, for the most part, well-written, with the authors' motivations and explanation of the method conceived quite straightforward.
2. The method has a convergence guarantee.
3. The authors also clearly emphasise their contribution and their novelty.
4. The results are convincing and show quite some promise.

Cons:

1. It would be better if the authors can provide some complexity analysis of the proposed algorithm and compared methods.
2. It would be better if the authors can add high-level intuition to help readers better understand the method section.

---

> ### Author Response · Authors · 2022-08-02
> **Response to Comments**
>
> We would like to thank the reviewer for the insightful comments. We have addressed the comments and accordingly modified the manuscript. Changes in the revised manuscript are highlighted in red color. Please find below our responses to comments:
>
> * *Complexity analysis*: We have added complexity analysis, see lines 214-222 of the revised version which is highlighted as red. We have added that “Each client needs to store $d$-dimensional $D$ random feature vectors for each kernel. Therefore, the memory requirement of each client to implement function approximation using POF-MKL is $\mathcal{O}(dND)$. Using POF-MKL, at each time instant, each client needs to perform $\mathcal{O}(dND)$ operations including inner products and summations. Furthermore, when $\xi_k < 1$, in order to choose a subset of kernels, the $k$-th client needs to sort kernels which imposes worst case computational complexity of $\mathcal{O}(N\log N)$. However, when $\xi_k=1$, according to PMF in equation (8), the $k$-th client chooses one bin uniformly at random and as a result in this case the $k$-th client does not need to sort kernels. Therefore, setting $\xi_k < 1$, the computational complexity for the $k$-th client is $\mathcal{O}(dND + N\log N)$ while setting $\xi_k=1$, the computational complexity of the $k$-th client at each time instant is $\mathcal{O}(dND)$.”
> * *High-level intuition*: We have added more intuitive explanations to section 3. Specifically, below equation (6) in line 157 of the revised version, we have added that “As it can be inferred from (6), each client constructs its own personalized combination of kernels.” Moreover, in lines 199-200 of the revised version we have added that “According to Algorithm 1, kernel subset selection is personalized since each client chooses its own subset of kernels to update their parameters.”
> * *Participation of subset of clients*: Theorem 1 does not require participation of all clients meaning that each client can achieve sub-linear regret with respect to the RF approximation of the best kernel even though a subset of clients participates in updating models at each time instant. However, participation of subset of clients instead of all clients makes the regret bound (14) in Theorem 2 looser. In other words, participation of subset of clients makes the regret of the server looser.
> * *Equation (7)*: Equation (7) is the update rule to update weights $w_{ik,t}$ and helps the proposed algorithm to achieve sub-linear regret. Each client $k$ assigns a weight $w_{ik,t}$ to the $i$-th kernel at time $t$ which indicates the confidence of the client $k$ to the predictions made by the kernel $i$. After making prediction using kernel $i$, the $k$-th client observes the label $y_{k,t}$. Then the $k$-th client computes the loss $\mathcal{L}(\hat f_{RF,it}(x_{k,t}),y_{k,t})$. Then in order to update the weight of $w_{ik,t}$ for the next time instant $t+1$, the $k$-th client multiply $w_{ik,t}$ by $\exp(-\eta_k \mathcal{L}(\hat f_{RF,it}(x_{k,t}),y_{k,t}))$ to obtain $w_{ik,t+1}$. Therefore, increase in the loss $\mathcal{L}(\hat f_{RF,it}(x_{k,t}),y_{k,t})$ leads to decrease in $w_{ik,t+1}$.

---

### Official Review · Reviewer_rNRS · 2022-07-11

**Rating:** 5
**Confidence:** 4
**Soundness:** 2 fair
**Presentation:** 2 fair
**Contribution:** 2 fair

**Summary:**

This paper considers the problem multi-kernel learning in online and distributed settings. Given a set of preselected kernel, clients should decide a **personalized** allocation (denoted $\mathbf{w}$ in the paper) of the kernels in an online fashion with the goal of minimizing their cumulative regret.  The paper leverage random feature (RF) approximation to solve the problem if ``curse of dimensionality'' usually encountered in online kernel learning. The RF approximation parameters (denoted $\theta$ in the paper) are estimated **globally** across clients. In order to reduce the communication costs, at   time step $t$, each client selects randomly a subset  of kernels  according  to  a probability mass function that aims to give more importance to kernels with larger allocation factor $\mathbf{w}_{ik, t}$.

The paper performs regret analysis, and proves that the each client achieves sub-linear regret with respect to the RF approximation of its best kernel in hindsight. Experimental results confirms the theoretical results and show the advantage of the proposed method in comparison to other online multi kernel federated learning methods.



**Questions:**

* For Theorem 2 to make sense, it is crucial to have a lower bound on $p_{i^{\*}k, t}$, or  to show that this term can not get arbitrary close to zero. I will raise my score if this issue is solved.

### Post rebuttal
The authors have partially addressed my concerns, I increase my score.

**Limitations:**

No the authors did not discuss negative societal impact of their work, and I do not think such discussion is needed.

**Strengths And Weaknesses:**

## Strengths

The paper is overall well written and easy to follow. The proposed idea is relevant and the theoretical results are correct.

## Weaknesses

### Originality and Related Work Discussion

**Reference [12] is not enough discussed.** &nbsp; The method proposed in [12] shares two similarities with POF-MKL. First, reference [12] uses RF approximation to handle  ``curse of dimensionality'' encountered in online kernel learning. Second, reference [12]  randomly  selects a subset of kernels to be communicated at every step in order to allow for a large  number  of  kernels to be  chosen  without  the  burden  of  communication  overhead. In this regard, this paper distinguishes itself from [12] by allowing clients to decide individually  the weights of each kernel, whereas, kernels' weights are global in [12]. Moreover the theoretical analysis is similar to the one performed in [12]. This discussion was not explicit in the paper and I think it will be useful if the similarities and differences with [12] were highlighted.

**Related Work.** &nbsp; Personalized federated learning is a well studied field that is closely related to this paper. However, only four references ([3, 6, 18, 31]) on personalized FL are cited, and none of them is discussed. As mentioned above, the main contribution of the paper (in comparison with [12]) is that it allows for kernel weights to be personalized for each client; it is therefore to discuss in more details  personalized FL literature. Moreover, the idea of learning personalized weights of globally learnt base functions was previously proposed in (Marfoq el al., 2021).

### Technical soundness

**Notation.** &nbsp; The notation is sometime misleading and makes some parts of the paper hard to follow. For example:

* The notation $\hat{f}_{RF, it}$   hides the dependence  on $\Theta$

* $\Theta_{t}$ is used to denote two different quantities: in line108, it is used as $\Theta_{t}=[\alpha_{1,1}, \dots, \alpha_{K,1}, \dots, \alpha_{1, t}, \dots, \alpha_{K,t}] \in \mathbb{R}^{tK}$, while in line 156, it is used as $\Theta_{t}=[\theta_{1, t}, \dots, \theta_{N,t}] \in \mathbb{R}^{D \times N}$

* Problem (1) should be formulated without $\Theta$ and $\mathbf{w}$ as $\min_{f\in \mathcal{H}}  \sum_{t=1}^{T}\sum_{k=1}^{K} \mathcal{L}(f(x_{k,t}), y_{k,t}) + \lambda \Omega(\left\||f\right\||^{2})$

**Theorem 2.** &nbsp; Although Theorem 2 is correct, it has few problems and can be improved. First,  The second term of the RHS of Eq. (14) is inversely proportional to $p_{i^{\*}k, t}$, which can be made arbitrary small when kernel $i^{\*}$ is not optimal for client k but is optimal for all other clients. Second, the coefficient $(1+\epsilon)$ in the first term of the RHS of Eq. (14) is not needed. This coefficient appears in Eq. (41) as a result of using the bound $\sup_{x, x'}|z_{i}^{\intercal}(x)z_{i}(x')| \leq 1 + \epsilon$; however one can simply use  $\sup_{x, x'}|z_{i}(x)^{\intercal}z_{i}(x')| \leq 1$, as the functions $\sin$ and $\cos$ are bounded by $1$. Finally, in the context of this work, I think it would be better to report the bound of the local regret defined with respect to the optimal local predictor at each client. This can be obtained as a corollary of Theorem 1 and Theorem 2.

P.S.: *The assumption that each data sample is bounded (line 227) is not needed.*


## References
Marfoq, O., Neglia, G., Bellet, A., Kameni, L., & Vidal, R. (2021). Federated multi-task learning under a mixture of distributions. Advances in Neural Information Processing Systems, 34, 15434-15447.

---

> ### Author Response · Authors · 2022-08-02
> **Technical soundness**
>
> **Theorem 2**
>
> * In equation (14), $p_{i^{\*}k,t}$ cannot be arbitrarily small. We specified the lower bound of $p_{i^\*k,t}$ in line 253 of the revised version (line 242 of the original submission). Please see line 201 of the revised version (line 199 of the original submission) where it is explained that $p_{ik,t} = q_{b_{ik},t}$ for all $i$  including $i^*$. According to equation (8), for any $j$, $k$ and $t$, $q_{jk,t} > \frac{\xi_k}{m}$ since for any $j$ and $k$, we have $u_{jk,t}>0$. Note that $u_{jk,t}$  is defined in line 190 of the revised version (line 189 of the original submission) as $u_{jk,t}= \sum_{\kappa_i \in B_j}{w_{ik,t}}$. Each weight $w_{ik,t}$ is initialized as 1 and updated according to equation (7). Therefore, the weights $w_{ik,t}$ are positive numbers and as a result weights $u_{jk,t}$ are positive numbers.  Therefore, we can conclude that for all $i$, $p_{ik,t}>\frac{\xi_k}{m}$ where $m=\lceil \frac{N}{M} \rceil \le N$. The value of $\xi_k$ is chosen by the proposed algorithm POF-MKL. Therefore, $p_{i^*k,t}$ is bounded from below and also its lower bound is controlled by the proposed algorithm to guarantee sub-linear regret. In the paper, we prove that for the proper choice of $\xi_k = \mathcal{O}(1), \forall k$, the proposed POF-MKL achieves sub-linear regret. Please see lines 252-261 in the revised version (lines 241-250 in the original submission).  Specifically, the exploration rate $\xi_k$ is chosen by the $k$-th client and the server can ask clients to choose exploration rates larger than a certain value.
> * We agree that the bound $\sup |z_i^\top(x)z_i(x^\prime)|\le 1+\epsilon$ is not necessary and can be improved to $\sup |z_i^\top(x)z_i(x^\prime)|\le 1$. Thanks to this comment, we have improved the regret bound in theorem 2. Also, we have revised the proof of Theorem 2. On the other hand, the assumption that each data sample is bounded is necessary for $\sup_{x,x^\prime} |z_i^\top(x)z_i(x^\prime) - \kappa_i(x,x^\prime)| \le \epsilon$ to be held true with probability greater than $1- 2^8 \left(\frac{\sigma_i}{\epsilon}\right)^2\exp\left(-\frac{D\epsilon^2}{4(d+2)}\right)$. See claim 1 in reference [35] of the revised manuscript for more detials. Such an assumption can be modified as  $\||x\|| \le C$ by introducing an additional constant, but we set it as 1 for simplicity of presentation.
> * In theorem 1, we prove that the regret of each client with respect to the RF approximation of the best kernel is sub-linear. In theorem 2, we prove that the regret of the server with respect to the best function approximator is sub-linear. The regret with respect to the best function approximator depends on kernel parameters $\{\theta_{i,t}\}_{i=1}^N$ . Since  parameters $\theta_\{i,t\}$ are learned globally over all data samples distributed among clients, it cannot be guaranteed that each client achieves sub-linear regret with respect to the best function approximator.
>
> **Notation**
> * We have revised $\hat f_{RF,it}(x_t)$ to $\hat f_{RF,it}(x_t;\theta_{i,t})$ to show dependence on parameter $\theta_{i,t}$.
> * We have revised the notation $\Theta_t = [\theta_{1,t},\ldots,\theta_{N,t}]$ to  $\hat\Theta_t = [\theta_{1,t},\ldots,\theta_{N,t}]$ throughout the paper.
> * Based on the discussion in lines 87 and 88, in problem (1), the server aims at learning the optimal parameters for the previously chosen function $f()$. Therefore, we did not revise the formulation of the problem (1).

---

> > ### Comment · Reviewer_rNRS · 2022-08-09
> > **Response to Authors**
> >
> > Thank you for you response. Your answers addresses part of my concerns. However I still do not agree on some of your answers.
> >
> > ### Theorem 2
> > Lowering bounding $p_{ik, t}$ using a lower bound on $\xi_{k}$ is a bit artificial. Also you say that $\xi_{k}$ is an exploration parameter; one can chose it arbitrary small. In regards of your answer, it seems that choosing $\xi_{k}=1$ is the best choice. If it is the case, then that puts into question the interest behind having $p_{ik, t}$ from the first place.
> >
> > ### Problem (1)
> > The current formulation of Problem (1) is misleading, when placed at the beginning of Section 2. With this formulation, you are obliged to talk about $\Theta$ and $w$ before properly defining them. I understand your choice of keeping the current formulation, but I really think it makes Section 2 harder to follow.

---

> > > ### Author Response · Authors · 2022-08-09
> > > **Response to Comment**
> > >
> > > Thank you so much for the discussion. Below we provide more explanations about the notation and exploration rate $\xi_k$ to resolve your remained concerns.
> > >
> > > **Exploration Rate**
> > >
> > > - Please note that $\xi_k$ is chosen by the $k$-th client and clients collaborate with the server to choose their personalized exploration rate such that the regret bound in (14) is of order $\mathcal O(\sqrt T)$. Specifically, the server determines the lower bound for exploration rate and the $k$-th client chooses exploration rate $\xi_k$ greater than the lower bound given by the server. Therefore, choosing exploration rate $\xi_k$ greater than a certain value determined by the server is beneficial for clients and there is no any reason that clients want to choose arbitrarily small values for $\xi_k$. Therefore, using the collaboration of clients and the server the regret bound of $\mathcal O(\sqrt T)$ is guaranteed.
> > >
> > > - This is true that $\xi_k = 1$ is the optimal choice in terms of regret bound. Note that $\xi_k=1$ refers to the case where the $k$-th client chooses a subset of kernels uniformly at random. However, in the paper we explain that choosing personalized value for $\xi_k$ promotes the privacy of clients since given the updated value for $\theta_{ik,t+1}$, the server can find $\nabla \mathcal L(\boldsymbol \theta_{i,t}^\top z_i(x_{k,t}),y_{k,t})/p_{ik,t}$ instead of the gradient of loss $\nabla \mathcal L(\boldsymbol \theta_{i,t}^\top z_i(x_{k,t}),y_{k,t})$. Note that $p_{ik,t}$ is chosen by the $k$-th client. Exchanging the gradients can be hazardous to the privacy of federated learning [52, 14]. Please see lines 209-213 in the revised version.
> > >
> > > To sum up, choosing the arbitrarily small value for $\xi_k$ can be hazardous to the performance of the $k$-th client. Therefore, the $k$-th client would not choose arbitrarily small $\xi_k$.
> > >
> > > **Notation of Problem (1)**
> > >
> > > Thank you for your feedback. We will revise the problem (1) based on your comment.

---

> > > > ### Comment · Reviewer_rNRS · 2022-08-09
> > > > **New comment**
> > > >
> > > > Thank you for your answer. I have two comments:
> > > >
> > > >
> > > > 1. Even if  client $k$ chooses $\xi_{k}=1$, the server can not immediately infer the value of $\nabla \mathcal{L}(\theta_{i,t}^{\intercal}z_{i}(x_{k,t}), y_{k,t})$ without access to the information that $\xi_{k}=1$
> > > >
> > > > 2. I understand from your previous comments that $\xi_{k}$ controls the privacy-utility trade-off of the algorithm. If it is the case, I think it could be beneficial to clearly discuss such a trade-off

---

> > > > > ### Author Response · Authors · 2022-08-09
> > > > > **Response to new Comment**
> > > > >
> > > > > Thank you so much for your insightful comments. Please find below our response to your comments:
> > > > >
> > > > > 1. This is true that if client $k$ chooses $\xi_k=1$, the server cannot be certain about the value of $\nabla \mathcal L(\boldsymbol \theta_{i,t}^\top z_i(x_{k,t}),y_{k,t})$ since the server does not know $\xi_k$. However, since $\xi_k=1$ is the best option in terms of regret, the server can assume that $\xi_k=1$ to estimate $\nabla \mathcal L(\boldsymbol \theta_{i,t}^\top z_i(x_{k,t}),y_{k,t})$. Therefore, if the client $k$ wants to make sure that the gradient $\nabla \mathcal L(\boldsymbol \theta_{i,t}^\top z_i(x_{k,t}),y_{k,t})$ is not revealed to the server, it can choose a value other than 1 for $\xi_k$.
> > > > >
> > > > > 2. We have added a discussion about privacy-utility trade-off to the revised version. We explained that choosing larger value for $\xi_k$ results in tighter regret bound and as a result the optimal choice for $\xi_k$ in terms of regret is $\xi_k=1$. However, choosing smaller values for $\xi_k$ makes the value of $p_{ik,t}$ more dependent on weights $w_{ik,t}$ (c.f. equation (8)). Therefore, choosing smaller values for $\xi_k$ makes $p_{ik,t}$ less predictable. This makes estimating $\nabla \mathcal L(\boldsymbol \theta_{i,t}^\top z_i(x_{k,t}),y_{k,t})$ given $\nabla \mathcal L(\boldsymbol \theta_{i,t}^\top z_i(x_{k,t}),y_{k,t})/ p_{ik,t}$ more difficult. This revision can be found in lines 262-267 of the latest revised version. Please note that choosing any $\xi_k < 1$, the server is not able to find $\nabla \mathcal L(\boldsymbol \theta_{i,t}^\top z_i(x_{k,t}),y_{k,t})$ given $\theta_{ik,t+1}$.
> > > > >
> > > > > Thank you again for your help to improve our paper.

---

> ### Author Response · Authors · 2022-08-02
> **Originality and Related Work Discussion**
>
> We would like to thank the reviewer for the insightful comments. We have addressed the comments and accordingly modified the manuscript. Changes in the revised manuscript are highlighted in red color. Please find below our responses to comments:
>
> **Discussion about reference [12]**:
>
> Based on this comment, we have extended the discussion about the comparison between the proposed POF-MKL and algorithms in [21] ([12] in the original submission). Revisions in the revised manuscript are highlighted as red. Revisions can be found in lines 282-303 of the revised manuscript. The following similarities and differences between the proposed POF-MKL and algorithms in [21] are highlighted in the extended discussion:
>
> Similarity with [21]: Both POF-MKL and algorithms in [21] exploit random feature approximation to alleviate computational complexity of online kernel learning. Furthermore, both POF-MKL and algorithms in [21] learn a linear combination of kernels.
>
> Differences with [21]:
> * The proposed POF-MKL allows clients to learn their own personalized combination of kernels (c.f. equation (6)).
> * Using the proposed algorithm, each client selects the subset of kernels to update them. Using the proposed algorithm, subsets of kernels selected for updating are different across clients. According to equation (8), each client is able to choose a subset of kernels based on the performance of kernels in previous rounds. Therefore, kernel selection is personalized using the proposed algorithm. However, in eM-KOFL of [21], at each time instant, the server selects one kernel for all clients. Personalized kernel selection provides the POF-MKL with more flexibility since using POF-MKL, even if each client chooses one kernel to update it,  it is possible that at every time instant, for each kernel, the server receives updates. However, using eM-KOFL of [21], at each time, the server only receives the update of one kernel. Therefore, using the proposed POF-MKL, the server would be able to learn the optimal parameter $\theta$ for each kernel faster than eM-KOFL of [21].
> * Moreover, personalized kernel selection brings challenges in regret analysis. In order to achieve sub-linear regret while clients perform personalized kernel selection, the update rule in equation (9) is proposed which is significantly different from update rules of algorithms in [21]. This differentiates the regret analysis of the proposed paper with regret analysis of [21]. Furthermore, according to Theorem 2 of [21], eM-KOFL achieves sub-linear regret with respect to the random feature approximation of the best kernel with probability $1-\delta$. Also, note that the regret in Theorem 2 of [21] is the cumulative regret of all clients. However, in Theorem 1 of the present paper, we prove that using the proposed POF-MKL, each client can achieve sub-linear regret with respect to the random feature approximation of its best kernel with probability 1 (i.e. there is no uncertainty in the regret upper bound unlike eM-KOFL of [21]). The personalized kernel selection and update rule proposed in equation (9) lead to these advantages compared with eM-KOFL of [21].
>
> **Related Work**
>
> We have added a discussion about personalized federated learning and cited more papers. Also, we compare previous works in personalized federated learning with the proposed POF-MKL in the added discussion. This revision can be found in lines 262-273 of the revised manuscript and is highlighted as red. Furthermore, we mention that in “Federated multi-task learning under a mixture of distributions” by Marfoq et al., (2021) a personalized model is a linear combination of a set of shared component models such that each client constructs its personalized mixture of models. Also we explained that previous works on personalized federated learning belong to offline cases and their theoretical guarantees cannot be applied to online learning tasks which is the main focus of this paper. In addition, previous personalized federated learning approaches require that clients store data samples in batch for training. However, the present paper assumes that clients may not be able to store observed data samples in batch. Using the proposed POF-MKL, clients do not need to store data samples.

---

### Official Review · Reviewer_KZ2t · 2022-07-11

**Rating:** 4
**Confidence:** 3
**Soundness:** 3 good
**Presentation:** 3 good
**Contribution:** 3 good

**Summary:**

This paper proposed a unified framework to train the online federated MKL model for clients efficiently. The model combined random feature approximation with federated learning and proposed a novel personalized online federated MKL algorithm (POF-MKL), which adds a new perspective for MKL models to achieve higher communication efficiency with a solid theoretical guarantee. Experiments are provided to validate the theoretical claims.

**Questions:**

It would be better to compare with more different online Federated MKL algorithms. For example, MK-OFL [1] also achieves the optimal sublinear regret bound and selects only one kernel for each client instead of a subset of kernels. What are the advantages when compared with MKOFL?

In line 276-277, there is no citation for each baseline, especially for OFSKL and OFMKL-Avg. Are they just two variations of the method Fed-OMD?

**Limitations:**

Limitation: see "Weakness".

negative societal impact: N/A

**Strengths And Weaknesses:**

Pros:

1. The proposed method applied random feature approximation to select a subset of kernels, which added a novel perspective to achieve high communication efficiency and real-time approximation.

2. Compared with existing online federated MKL algorithms (eM-KOFL and vM-KOFL), this paper proposed a personalized federated MKL model based on a more scalable and adaptive pattern, where each client chose a subset of kernels and sent partial updates according to clients-server
communication bandwidth.

Cons/Questions:

1. The presentation in the experiments section is somewhat not well-structured. The writing could be improved by re-framing the description of results, giving the readers a more intuitive explanation, and reducing the citation of “Table 1”.

2. It would be interesting to see the performance of the real online learning tasks such as online regressions and time-series predictions.

3. It would be more convincing to provide the convergence time/speed of the proposed algorithm.

4. It would be better to compare with more different online Federated MKL algorithms. For example, MK-OFL [1] also achieves the optimal sublinear regret bound and selects only one kernel for each client instead of a subset of kernels. What are the advantages when compared with MKOFL?

[1] Chae, J., & Hong, S. (2021). Multiple Kernel-Based Online Federated Learning. arXiv preprint
arXiv:2102.10861.

5. In line 276-277, there is no citation for each baseline, especially for OFSKL and OFMKL-Avg. Are they just two variations of the method Fed-OMD?

---

> ### Author Response · Authors · 2022-08-02
> **Response to Comments**
>
> We would like to thank the reviewer for the insightful comments. We have addressed the comments and accordingly modified the manuscript. Changes in the revised manuscript are highlighted in red color. Please find below our responses to comments:
>
> 1- *The presentation in the experiments section*:
>
> We revised the descriptions of results. We have reduced the number of citations of Table 1. Furthermore, we have added more intuitive explanations about the reasons that the proposed POF-MKL performs better than other baselines. We have highlighted these added explanations as red in the revised version.
>
> * In order to explain the reason that POF-MKL with $M=25$ and $D=20$ works better than POF-MKL with $M=51$ and $D=9$, in lines 349-351 we have added that “Note that according to Theorem 2, increase in $D$ increases the probability that the server achieves sub-linear regret with respect to the best function approximator.”
>
> * Furthermore, to compare POF-MKL and vM-KOFL, in lines 354-356, we have added that “In fact, the optimal combination of kernels can be different across clients. Using POF-MKL, each client constructs its own personalized combination of kernels which results in lower MSE compared to vM-KOFL.”
>
> * Moreover, to compare POF-MKL with OFSKL, in lines 364-368, we have added that “utilizing multiple kernels enables POF-MKL to obtain lower MSE than that of OFSKL. In fact, using POF-MKL clients learn a linear combination of kernels which is proved to enjoy sub-linear regret with respect to the best kernel in hindsight while employing OFSKL clients have to make predictions using a pre-selected kernel.”
>
> 2- *It would be interesting to see the performance of the real online learning tasks such as online regressions and time-series predictions.*
>
> In experiments, we tested the performance of algorithms on online regression tasks. In the first line of section 4, we mentioned that the performance of the proposed POF-MKL is tested on online regression. In order to perform the experiments, we chose regression datasets. At each time instant, a new data sample which has not been observed by any clients is given to each client and the client makes a prediction for the newly observed data sample. Then the client incurs a loss and based on the observed loss, the client updates parameters. Note that clients do not store observed data samples to use in the future time instants. Furthermore, Air dataset [50] (reference [35] of the original submission) is a time-series dataset.
>
> 3- *the convergence time/speed of the proposed algorithm*
>
> In Theorems 1 and 2, it is proved that when the time horizon $T$ is reached, the proposed POF-MKL achieves sub-linear regret of $\mathcal{O}(\sqrt{T})$. In line 232 and 244 of the original submission, the order of the regret bounds are obtained as $\mathcal{O}(\sqrt{T})$. This shows the convergence rate of the algorithm. Also, it is useful to mention that online learning tasks should be continued until the time that the learner makes decision for all data samples that it observes and online learning tasks are not terminated when a certain convergence is achieved (see e.g. [R1], [R2] and [R3]). The present paper studies online learning tasks. Usually, online learning algorithms are evaluated based on the order of the regret bound up until the time horizon $T$ (see e.g. [R1], [R2] and [R3]). Theorems 1 and 2 in the present paper, obtain the regret bound up until the time horizon $T$.
>
> 4- *Comparison with MK-OFL of "Multiple Kernel-Based Online Federated Learning"*:
>
> We did compare the performance of the proposed algorithm with the mentioned reference. “Multiple Kernel-Based Online Federated Learning” is the arxiv version of [12] ([21] in the revised version of the paper). In [12] ([21] of the revised version), MK-OFL in the arxiv version  is called eM-KOFL, and we compared the proposed algorithm with the most updated and recent version.
>
> 5- *Citations of Baselines*:
>
> OFSKL and OFMKL-Avg are two variations of Fed-OMD. In the revised version, in lines 310, we added that “OFSKL and OFMKL-Avg are two variations of Fed-OMD [33].” We have cited all baselines in the beginning of section 4. Please see the lines 306 and 307 in the revised version. Revisions are highlighted as red.
>
> **References**
>
> [R1] N. Alon, N. Cesa-Bianchi, O. Dekel, and T. Koren, “Online learning with feedback graphs: Beyond bandits,” in Proceedings of Conference on Learning Theory, vol. 40, (Paris, France), pp. 23–35, Jul 2015.
>
> [R2] N. Alon, N. Cesa-Bianchi, C. Gentile, S. Mannor, Y. Mansour, and O. Shamir, “Nonstochastic multi-armed bandits with graph-structured feedback,” SIAM Journal on Computing, vol. 46, no. 6, pp. 1785–1826, 2017.
>
> [R3] T. Kocak, G. Neu, and M. Valko, “Online learning with Erdos-Renyi side-observation graphs,” in Proceedings of Conference on Uncertainty in Artificial Intelligence, p. 339–346, Jun 2016.

---

> > ### Comment · Reviewer_KZ2t · 2022-08-09
> > **Post Rebuttal Thoughts**
> >
> > Thanks for the authors' response and clarification. Part of my concerns have been addressed pretty well. While for the convergence time/speed, personally I still believe presenting some empirical results (in addition to theoretical analysis) will be beneficial and necessary.

---

> > > ### Author Response · Authors · 2022-08-09
> > > **Response to Post Rebuttal Thoughts**
> > >
> > > Thank you so much for your comment and taking time to review our paper. In Table 1 of section 4, we reported the run time of algorithms. The reported run time refers to average total run time of clients to perform online learning task on the entire data samples that they observe. Therefore, reported run time shows how much time it takes for clients to achieve the guaranteed sub-linear regret. Hence, the run times in Table 1 indicate the convergence time of algorithms.

---

### Meta-Review · Area_Chair_AKwR · 2022-08-27

**Recommendation:** Accept
**Confidence:** Less certain

**Metareview:**

The authors carefully designed the algorithm and presented their results.
The reviewers commented on the analysis of running costs, discussion on related works, and experimental comparisons.
The authors, as far as I can see, have addressed these appropriately.
The reviewers did not respond to the updates, but the scores should be increased, and I have included them in my decision.

**Award:**

No

---

### Decision · Program_Chairs · 2022-09-14

Accept